# Pericytes control vascular stability and auditory spiral ganglion neuron survival

**Yunpei Zhang[1†], Lingling Neng[1†], Kushal Sharma[1], Zhiqiang Hou[1], Anatasiya Johnson[1], Junha Song[2], Alain Dabdoub[3,4,5], Xiaorui Shi[1]\***

[1]Oregon Hearing Research Center, Department of Otolaryngology/Head & Neck Surgery, Oregon Health & Science University, Portland, United States; [2]Life Sciences Division, Lawrence Berkeley National Laboratory, Berkeley, United States; [3]Biological Sciences, Sunnybrook Research Institute, Toronto, Canada; [4]Department of Otolaryngology-Head & Neck Surgery, University of Toronto, Toronto, Canada; [5]Department of Laboratory Medicine and Pathobiology, University of Toronto, Toronto, Canada

**Abstract** The inner ear has a rich population of pericytes, a multi-functional mural cell essential for sensory hair cell heath and normal hearing. However, the mechanics of how pericytes contribute to the homeostasis of the auditory vascular-neuronal complex in the spiral ganglion are not yet known. In this study, using an inducible and conditional pericyte depletion mouse (PDGFRB-CreER[T2]; ROSA26iDTR) model, we demonstrate, for the first time, that pericyte depletion causes loss of vascular volume and spiral ganglion neurons (SGNs) and adversely affects hearing sensitivity. Using an in vitro trans-well co-culture system, we show pericytes markedly promote neurite and vascular branch growth in neonatal SGN explants and adult SGNs. The pericyte-controlled neural growth is strongly mediated by pericyte-released exosomes containing vascular endothelial growth factor-A (VEGF-A). Treatment of neonatal SGN explants or adult SGNs with pericyte-derived exosomes significantly enhances angiogenesis, SGN survival, and neurite growth, all of which were inhibited by a selective blocker of VEGF receptor 2 (Flk1). Our study demonstrates that pericytes in the adult ear are critical for vascular stability and SGN health. Cross-talk between pericytes and SGNs via exosomes is essential for neuronal and vascular health and normal hearing.

**\*For correspondence:**
shix@ohsu.edu

[†]These authors contributed equally to this work

## Editor's evaluation

This study identifies the roles of the pericytes in maintaining vascular volume and integrity of spiral ganglion neurons (SGNs) in the cochlea, the main hearing organ. It demonstrates that the roles are achieved mainly through the interactions between pericyte-released exosomes containing VEGF-A and VEGFR2-expressing the vessels and SGNs. Understanding the roles of organ-specific pericytes is paramount, making this study timely and significant. The study would be interesting for biomedical biologists working on hearing, blood vessels, signaling, and cell-to-cell interactions.

## Introduction

The inner ear is dense in vascular beds. Microvascular networks are situated in different locations of the cochlea (*Shi, 2011*). The major microvascular network is located in the cochlear lateral wall, receiving ~80% of cochlear blood flow (*Gyo, 2013*). Blood flow to the cochlear lateral wall is essential for cochlear homeostasis and is particularly important for generating the endocochlear potential necessary for sensory hair cell (HC) transduction (*Hibino et al., 2010*). The next largest microvascular network is situated in the region of the spiral ganglion neurons (SGNs), comprising ~19–24% of

cochlear blood flow. The network is critical for neural activity (*Angelborg et al., 1984*; *Gyo, 2013*; *Nakashima et al., 2001*), in particular, the vascular network of the spiral ganglion directly delivers nutrients and growth hormones to SGNs (as illustrated in *Appendix 1—figure 1*). Defects in vascular structure and function would significantly and quickly affect the viability of vulnerable neural tissue. Loss of SGNs is commonly associated with different types of hearing loss (*Leake et al., 2020*; *Viana et al., 2015*).

Pericytes are a type of mural cell that encircle endothelial cells of capillaries, pre-capillary arterioles, and post-capillary venules (*Armulik et al., 2011*). They extend their processes along and around these vessels and play a critical role in regulating vascular morphogenesis and function (*Attwell et al., 2016*; *Birbrair, 2018*). Normal function of pericytes and adequate blood supply is crucial for providing vital oxygen, ions, and glucose to meet organ metabolic needs (*Shaw et al., 2018*). Pericyte degeneration or loss has been identified as a major pathology in many diseases, including ischemic stroke (*Greif and Eichmann, 2014*), brain trauma (*Zehendner et al., 2015*), myocardial infarction (*O'Farrell and Attwell, 2014*), diabetic retinopathy (*Pfister et al., 2008*), and neurodegeneration diseases such as Alzheimer's disease, Parkinson's disease, and Huntington's disease (*Sweeney et al., 2018*). Pericytes are known to display phenotypic and functional heterogeneity, which is highly associated with organ function (*Dias Moura Prazeres et al., 2017*). While little is known about the role of pericytes in the inner ear; nothing is known about how pericyte loss affects peripheral neuron health.

SGN cell bodies, located in Rosenthal's canal, extend distal processes radially outward into the spiral lamina toward sensory HCs in the organ of Corti and central processes project into the auditory nerve (as illustrated in *Appendix 1—figure 1*; *Nayagam et al., 2011*). The microvascular network, located in the spiral ganglion region, forms radial vascular twigs that directly supply nutrients to the SGNs. Although the volume of blood flow to the spiral lamina and SGN is lower than to the cochlear lateral wall, it is critical for neuronal activity. Constant sound stimulation to the inner ear imposes a high energy demand on neurons, requiring rapid delivery of oxygen and glucose. Previously, we demonstrated the microvascular network in the spiral ganglion region is richly populated by pericytes (*Jiang et al., 2019*). Pericytes actively communicate with SGNs as evidenced by the pericyte-released particles observed in the soma of SGNs (*Jiang et al., 2019*). However, the role of pericytes in SGN viability and stability of vascular beds is not yet known. In this study, we use an inducible and conditional genetic pericyte ablation model, in combination with two newly established co-culture models, to demonstrate that pericyte loss in vivo causes loss of vessel volume and SGNs in adult mice. In vitro, we demonstrate vigorous vascular and neuronal growth in the neonatal SGN explants in the presence of exogenous pericytes. Increased SGN survival and neurite growth were also observed in adult SGNs. Most interestingly, we find the promotion of vascular and neuronal growth is mediated by pericyte-released exosomes, nano-sized 50–150 nm extracellular vesicles (EVs), containing vascular endothelial growth factor-A (VEGF-A). VEGF-A binding to VEGFR2 (Flk1) expressed in recipient cells, including endothelial cells and SGNs, controlled vascular and neuronal growth. Our data demonstrate for the first time that loss of pericytes leads to a reduction in vascular density and degeneration of SGNs, underscoring the vital role of pericytes in vascular and auditory peripheral neural health.

## Results

### The vascular network in the spiral ganglion region is rich in pericytes, and there is cross-talk between pericytes and SGNs

Normal capillary blood flow is controlled by pericytes. Pericytes are specialized mural cells, which surround the endothelial cells of small blood vessels and are vital for normal vascular function. Pericyte pathology, such as pericyte loss or damage, is a significant factor in degenerative neural diseases, including aging-related hearing loss (*Neng et al., 2015*), Alzheimer's disease, and brain dementia (*Miners et al., 2019*). The cochlear microvasculature in the spiral ganglion region contains a rich population of pericytes, as illustrated in *Figure 1a*. Using neural-glial antigen 2+ (NG2+) pericyte fluorescent reporter mice (illustration *Figure 1b*) in combination with fluorescence conjugated lectin blood vessel labeling and β III tubulin immunofluorescence labeling of SGNs, we visualized the vascular structure, pericyte distribution, and pericyte relationship with SGNs. Pericytes are predominantly distributed in the vascular bed, as shown in *Figure 1A*. Most interestingly, in NG2+ pericyte fluorescent reporter mice, we observed NG2-positive particles (likely released by pericytes) in

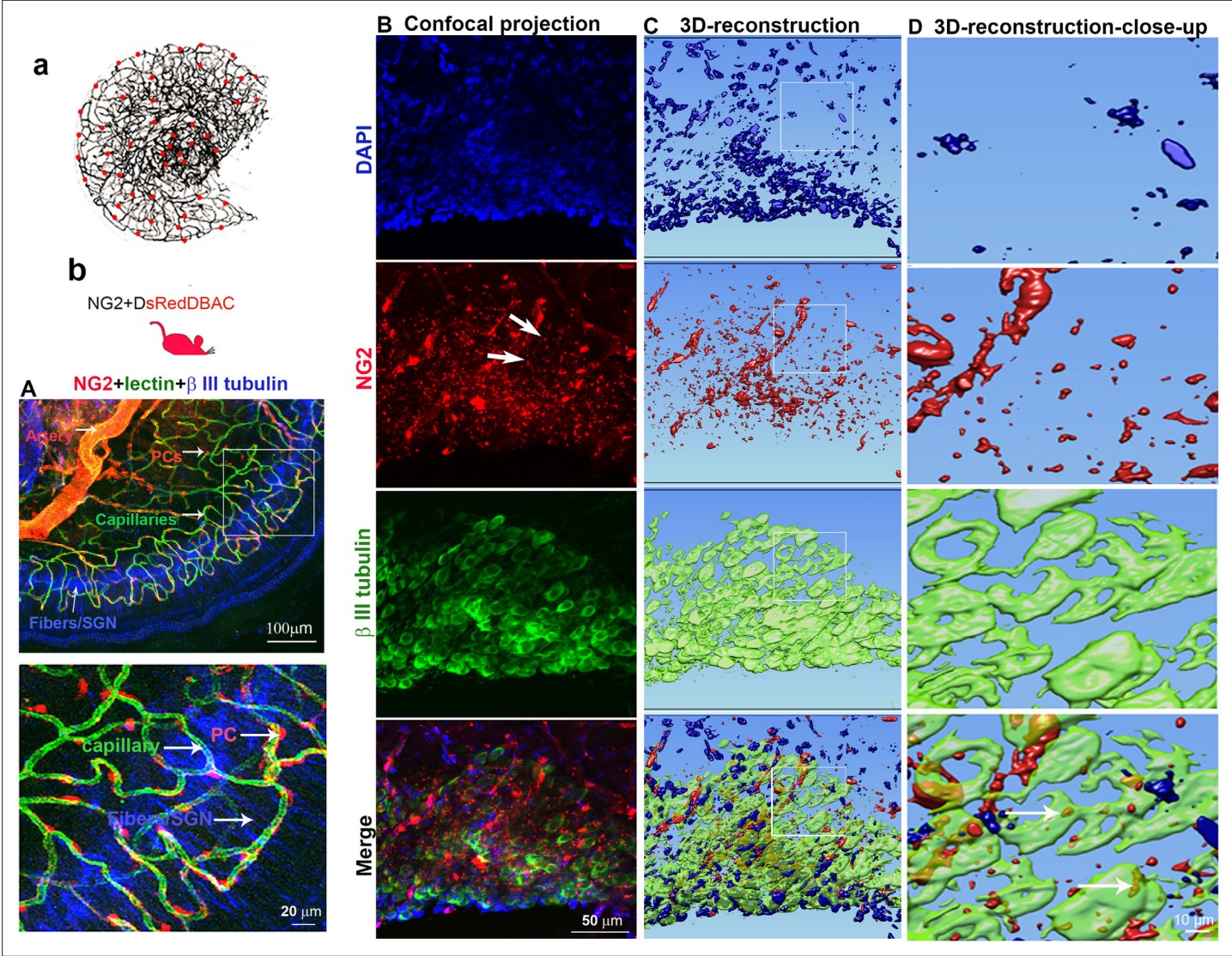

**Figure 1.** Cochlear vascular networks in the spiral ganglion region are densely populated by pericytes, and spiral ganglion neurons (SGNs) contains particles released by pericytes. (**a and b**) Illustrations of pericyte containing microvascular networks in the spiral ganglion region. (**A**) A confocal projection image of the spiral lamina from an NG2DsRedBAC mouse shows pericytes (red) situated on microvessels labeled with Lectin-Alexa Fluor 488 conjugate (green) around SGNs and their peripheral fibers labeled with an antibody for β-III tubulin (blue). The pericyte distribution can be better visualized under high magnification (A, lower). (**B–D**) Cross-talk between SGNs and pericytes is suggested by the red fluorescent particle (arrows) observed in SGNs (green). Further evidenced by 3D reconstructive images showing the particles are inside the soma of SGNs.

the soma of SGNs (*Figure 1B* lowest panel). The events are clearly visualized in 3D reconstructed images (*Figure 1C*, lowest panel) and magnified images, see *Figure 1D* lowest panel. This observation is consistent with our previous observation (*Jiang et al., 2019*), suggesting active communication between the pericytes and neural system. Substances released by pericytes may be essential for SGN health and pathology.

### Loss of pericytes leads to reduced capillary volume in the spiral ganglion region in vivo

To investigate whether pericyte loss affects vascular stability in the spiral ganglion of adult mice, we created an inducible pericyte depletion mouse model (PDGFRB-CreER$^{T2}$; ROSA26iDTR) by crossing PDGFRB-CreER$^{T2+/-}$ transgenic mice with ROSA26iDTR$^{+/+}$ mice, an inducible diphtheria toxin receptor (iDTR) mouse line carrying a Cre-dependent simian DTR, which leads to cell death with the administration of the diphtheria toxin (DT; *Buch et al., 2005*; *Zhang et al., 2021*; as illustrated in *Figure 2A*).

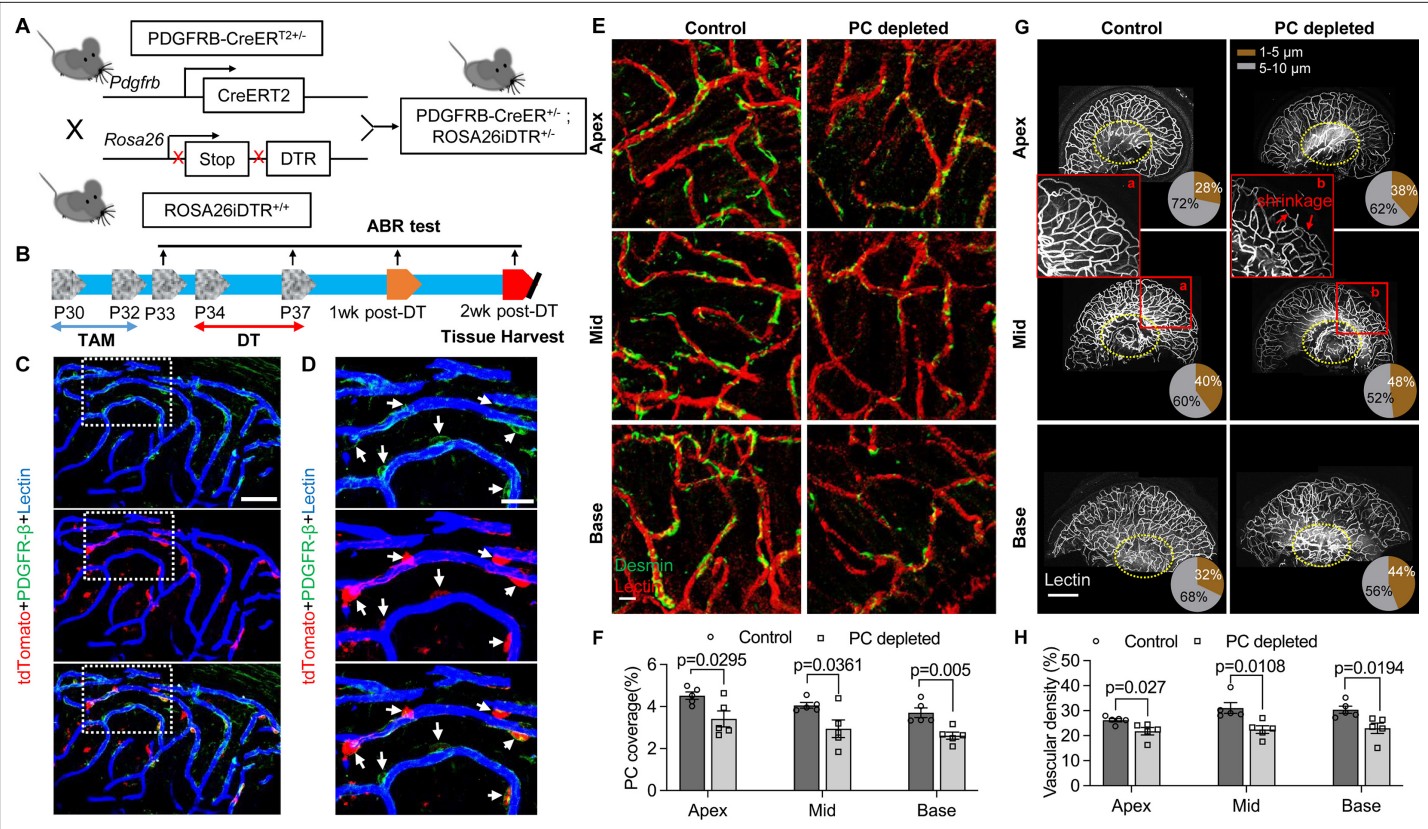

**Figure 2.** Pericyte-depletion induces vascular regression in the region of spiral ganglion neuron (SGN) peripheral nerve fibers in adult mice. (**A**) Schematic of a pericyte-depletion mouse model incorporating an inducible Cre-loxP system. (**B**) The diagram shows the timeline of tamoxifen and diphtheria toxin (DT) administration and the time point of auditory brainstem response (ABR) test and tissue harvest. (**C**) Co-localization of PDGFRβ-Cre (tdTomato) and immune-labeled PDGFRβ (green) signals in the pericytes of PDGFRB-CreER$^{T2}$; ROSA26tdTomato mice. (**D**) A high-magnification image further shows co-localization of the Cre and PDGFRβ fluorescence signals (arrows). (**E**) Representative figures show pericyte coverage in the region of the spiral limbus in DT-treated control inducible diphtheria toxin receptor (iDTR; left) and DT-treated PDGFRB-CreER$^{T2}$; ROSA26iDTR mice (right) 2 wk after DT injection. DT injection significantly leads to loss of pericyte coverage. (**F**) Pericyte density was significantly reduced in the PDGFRB-CreER$^{T2}$; ROSA26iDTR mice at the apical, middle, and basal turn relative to density in the control of iDTR mice (n=5, $p_{Apex}$=0.0295, $p_{Mid}$=0.0361, $p_{Base}$=0.005, unpaired t test). (**G**) Representative figures show the capillaries of the spiral lamina in control ROSA26iDTR (left) and PDGFRB-CreER$^{T2}$; ROSA26iDTR (right) mice, with the distribution of vessel diameter shown in π charts, and the location of SGNs shown in ellipses. (**H**) Total vascular density in the spiral limbus and lamina is significantly reduced in the pericyte-depleted mice 2 wk after DT injection (n=5, $p_{Apex}$=0.027, $p_{Mid}$=0.0108, $p_{Base}$=0.0194, unpaired t test). Loss of vascular volume with pericyte depletion is better seen in the high-magnification image inserts in panel G (a) and (b). Data are presented as the mean ± SEM. Scale bars: C, 50 µm; D, 20 µm; E, 20 µm; F, 200 µm.

This PDGFRB-CreER$^{T2+/-}$; ROSA26iDTR$^{+/-}$ mouse line at 1 mo of age received tamoxifen (TAM) for 3 d to induce the expression of Cre recombinase. To ablate pericytes, DT was administrated by daily intraperitoneal injection at 10 µg/kg for four consecutive days beginning 1 d after TAM (**Zhang et al., 2021**; as illustrated in **Figure 2B**). The PDGFRB-CreER$^{T2-/-}$; ROSA26iDTR$^{+/-}$ mice from same litters of PDGFRB-CreER$^{T2+/-}$; ROSA26iDTR$^{+/-}$ mice were treated with TAM and DT and constituted a control group. The cellular location of Cre recombinase expression under the *pdgfrb*-promotor was confirmed by crossing the PDGFRB-CreER$^{T2}$ mice with ROSA26tdTomato mice. As expected, the tdTomato fluorescence signal (**Greenhalgh et al., 2013**) largely co-localized with the immunofluorescence signal for the pericyte marker protein, PDGFRβ (green) in the vascular networks, as shown in **Figure 2C and D**.

Our results showed, relative to control mice, pericyte distribution (labeled with desmin) in the vascular networks (labeled with Alexa Fluor 649 conjugate lectin, a widely used fluorescence dye to visualize blood vessels [**Meyer et al., 2008**]), markedly reduced in the spiral ganglion regions of all cochlear turns 2 wk after DT treatment in the CreERT2/iDTR mice (n=5, $p_{Apex}$=0.0295, $p_{Mid}$=0.0361, $p_{Base}$=0.005, unpaired t-test; **Figure 2E and F**). In addition, the vascular volume in the spiral ganglion region was also notably reduced. **Figure 2G** has representative confocal images showing the pattern

of blood vessel distribution in control and pericyte-depleted groups. *Figure 2H* shows total vascular density in the spiral lamina is significantly reduced in the pericyte-depleted mice 2 wk after DT injection (n=5, $p_{Apex}$=0.027, $p_{Mid}$=0.0108, $p_{Base}$=0.0194, unpaired t-test). In addition, vascular shrinkage, a sign of degeneration, was frequently observed (as highlighted under high magnification, *Figure 2G*). Our results clearly indicate that pericytes are essential for vascular stability.

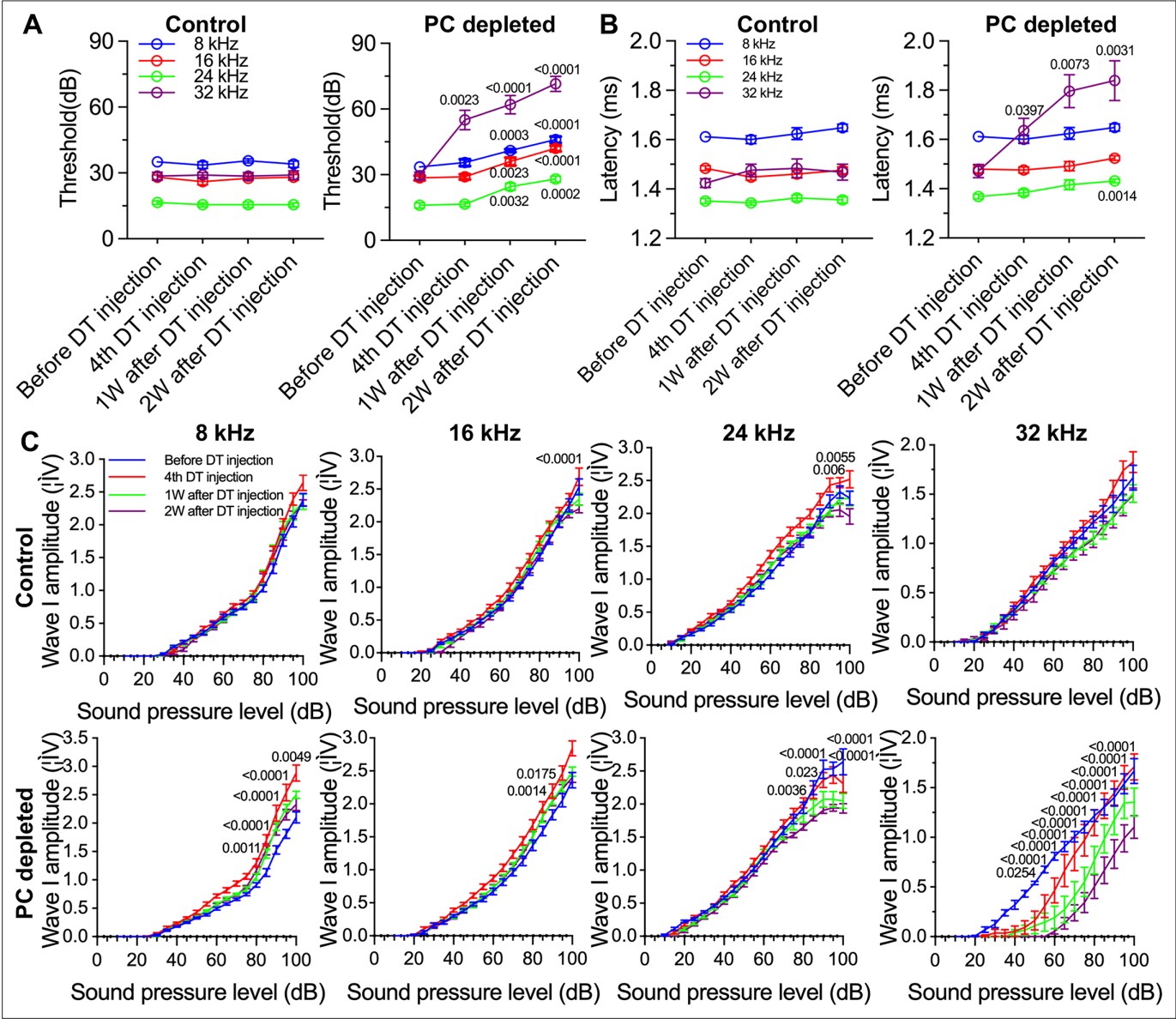

**Figure 3.** The depletion of pericytes led to hearing loss, increased latency and reduced amplitude of wave I. (**A**) The control mice showed no significant hearing threshold change after diphtheria toxin (DT) injection (n=10, p=0.9651). In contrast, the hearing threshold in pericyte-depleted animals was significantly elevated at 1 wk after DT injection and persisted to 2 wk after injection (n=10, p<0.0001). Two-way ANOVA, followed by Dunnett's multiple comparison test, and individual p values of different time points versus before DT injection are labeled on the graph. (**B**) The control mice showed no significant wave I latency change after DT injection (n=10, p=0.3576). In pericyte-depleted animals, the latency was significantly delayed at high frequency (32 kHz) starting with the fourth DT injection, and at low frequency (8 kHz) 2 wk after DT injection (n=10, p<0.0001). Two-way ANOVA, followed by Dunnett's multiple comparison test, and individual p values of different time points versus before DT injection are labeled on the graph. (**C**) Although changed wave I amplitude was randomly observed in control mice after DT injection, changes rarely persisted to 2 wk after the injection (n=10, $p_{16kHz}$<0.0001, $p_{24kHz}$<0.0001). In contrast, significant reduction in wave I amplitude was observed in pericyte-depleted animals at 2 wk after DT injection, particularly at high frequency (32 kHz) (n=10, $p_{8kHz}$<0.0001, $p_{16kHz}$<0.0001, $p_{24kHz}$<0.0001, $p_{32kHz}$<0.0001). Two-way ANOVA, followed by Dunnett's multiple comparison test, and individual p values at 2 wk after DT injection versus before DT injection at different sound pressure levels are labeled on the graph. Data are presented as the mean ± SEM.

## Pericytes are critical for hearing sensitivity, as well as for SGN function and survival in vivo

We further assessed hearing function by auditory brainstem response (ABR) in pericyte-depleted mice. As shown in *Figure 3A*, we found the control mice showed no significant hearing threshold change after DT injection. In contrast, the hearing threshold in pericyte-depleted animals was significantly elevated at 1 wk after DT injection and persisted to 2 wk after DT injection (n=10, p<0.0001, two-way ANOVA), consistent with our previous observation (*Zhang et al., 2021*).

In addition to the increased hearing threshold, we specifically examined ABR wave I, including its latency and amplitude, which is generally accepted as an indicator of auditory nerve activity in most mammalian animals and humans (*Xie et al., 2018*). We found the control mice showed no significant wave I latency change after DT injection. In contrast, the wave I latency in pericyte-depleted animals was significantly delayed, beginning at high frequency (32 kHz, after fourth DT injection) and extending to lower frequency (8 kHz) at 2 wk after DT injection (n=10, p<0.0001, two-way ANOVA; *Figure 3B*). Correspondingly, the significantly decreased amplitude of wave I was observed at all frequencies but started at much lower sound pressure level (SPL; dB) at high frequency (32 kHz) 2 wk after DT injection (n=10, p<0.0001, two-way ANOVA; *Figure 3C*). Although a change in wave I amplitude was occasionally observed in control mice after DT injection, the change rarely persisted to 2 wk after the DT injection.

In our pericyte-depleted animals, we observed significant SGN loss at all turns (*Figure 4A and B*; n=9, $p_{Apex}$=0.0085, $p_{Mid}$=0.0099, $p_{Base}$=0.0127, unpaired t-test) with markedly decreased β-III tubulin expression 2 wk after DT treatment at the middle and basal turns (*Figure 4A and C*; $p_{Apex}$=0.1074, $p_{Mid}$<0.0001, $p_{Base}$<0.0001, unpaired t-test). β-III tubulin, a specific marker for neuronal cytoskeleton, is commonly used to distinguish between different types of neurons (*Katsetos et al., 2003*). Reduced tubulin is an indication of structural disassembly seen in many neurodegenerative diseases and taken as a sign of neuronal dysfunction (*Baas et al., 2016*). These results suggest loss of pericytes affects the viability of the spiral ganglion in adults.

## Transcriptome analysis of cochlear pericytes

To gain a better understanding of the properties of cochlear pericytes, we investigated the transcriptome of primary cochlear pericytes cultured from postnatal (P10-P15) C57BL/6J mice. This cell line was generated from the stria vascularis by a well-established 'mini-chip' protocol as previously described (*Neng et al., 2013a*), and passages 3–6 were used. Total RNA isolated from cultured cochlear pericytes was subjected to RNA-sequencing (RNA-seq) analysis. 12887 genes (RPKM (reads per kilobase of exon per million reads mapped)>0) were identified expressed in mouse cochlear pericytes, 7889 (RPKM >0.5) of which were further analyzed by performing an overrepresentation test of Protein Analysis Through Evolutionary Relationships (PANTHER) pathways (relative to the whole-genome for *Mus musculus*) in the PANTHER classification system (version 17.0; *Mi et al., 2013*; *Mi et al., 2019*). We identified 59 overrepresented PANTHER pathways in our cochlear pericyte dataset (*Table 1*), including angiogenesis related ('CCKR signaling map,' P06959; 'Angiogenesis,' P00005; 'VEGF signaling pathway,' P00056) and neurodegeneration related ('Alzheimer disease-presenilin pathway,' P00004; 'Alzheimer disease-amyloid secretase pathway,' P00003; '5HT2 type receptor mediated signaling pathway,' P04374) pathways among the top 20 statistically significant results (*Figure 5A*).

## Pericytes promote vascular and neuronal growth in the spiral ganglion in vitro

We next investigated whether pericytes directly communicate with nearby cells and affect the corresponding vascular and neuronal biology. Neonatal SGN explants and adult SGNs were cultured with exogenous pericytes in a transwell co-culture system, as illustrated in *Figure 5B and D*. After 5 d in culture, we observed numerous and longer dendritic fibers, as well as new vascular sprouting, in the exogenous pericyte treated SGN explants relative to controls (*Figure 5B*). There were significant differences in the number of new vessel branches and in number and length of dendritic fibers between the two groups (n=6, $p_{Vascular\ branch\#/area}$=0.0018, $p_{Neurites\#/area}$<0.0001, $p_{Neurite\ length/explant}$=0.0393, unpaired t test; *Figure 5C*). The exogenous pericytes also consistently promoted survival and new neurite growth in adult SGNs (*Figure 5D*). Significant differences in cell survival and in average neurite number and length were observed between the two groups (n=3 wells, 25 cells per well,

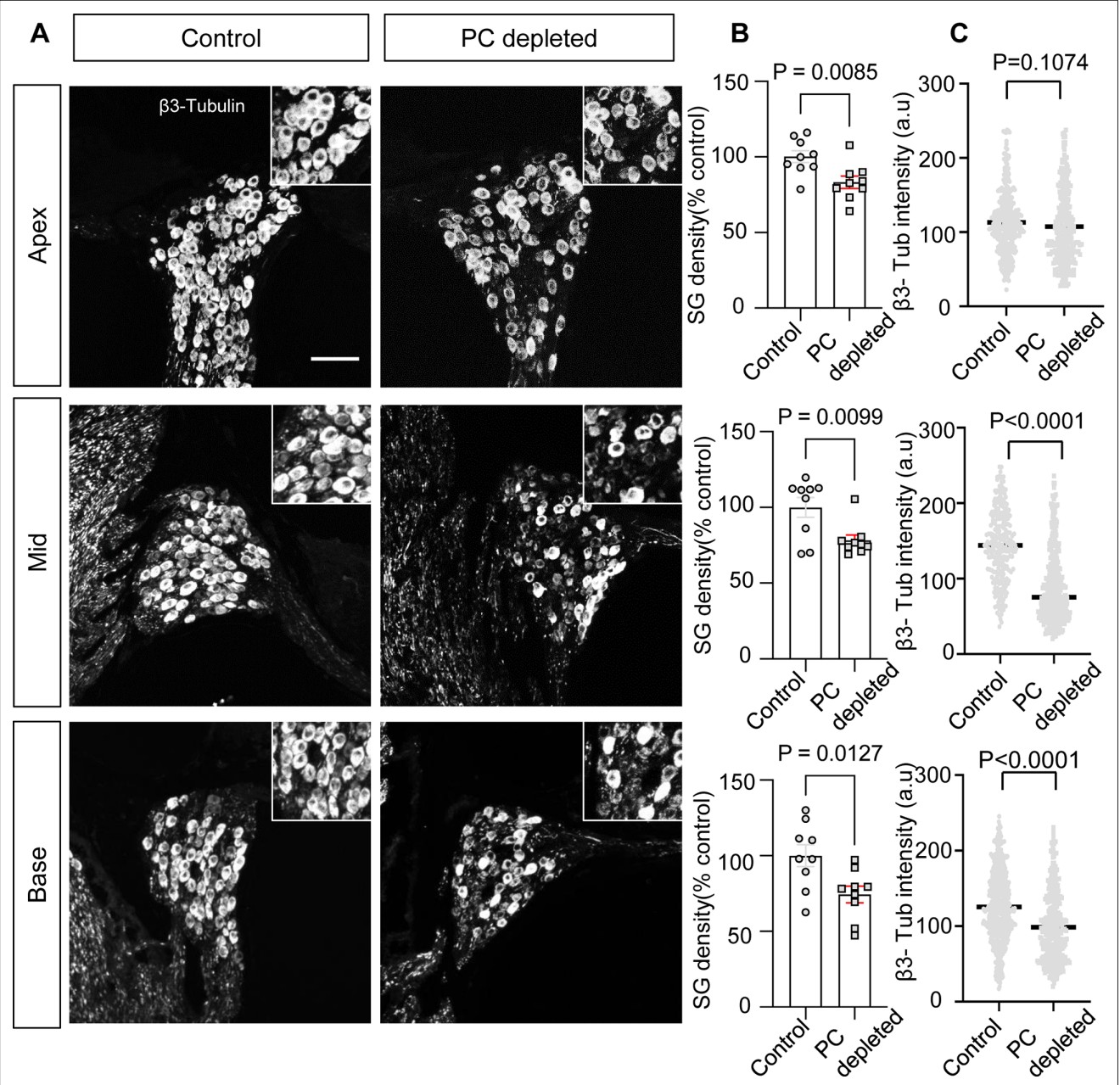

**Figure 4.** The depletion of pericytes led to spiral ganglion neuron (SGN) loss and decreased expression of β-III tubulin in SGNs. (**A**) Representative confocal images from control and pericyte-depleted animals, labeled with antibody for β-III tubulin. (**B**) Significant SGN loss at all turns 2 wk after diphtheria toxin (DT) treatment (n=9, $p_{Apex}$=0.0085, $p_{Mid}$=0.0099, $p_{Base}$=0.0127, unpaired t-test). (**C**) Significantly decreased β-III tubulin expression in SGNs at middle and basal turns 2 wk after DT treatment ($p_{Apex}$=0.1074, $p_{Mid}$<0.0001, $p_{Base}$<0.0001, unpaired t-test). Data are presented as the mean ± SEM. Scale bar: D, 50 μm.

$p_{neuron\ survival}$=0.0186, $p_{Neurites\#/cell}$<0.0001, $p_{longest\ neurite\ length/cell}$<0.0001, unpaired t test; *Figure 5E*). The data clearly indicate that pericytes mediate both vascular and neural growth through extracellular communication in both neonatal and adult SGN tissue, as well as demonstrate that pericytes are essential for vascular and neuronal growth both during development and in the adult.

## Identification of cochlear pericyte-derived exosomes

We then asked how pericytes communicate with neighboring cells. Many cells secrete exosomes, carrying cargo including all known molecular constituents of the host cell, including protein, DNA,

**Table 1.** Overrepresented PANTHER pathways for genes (RPKM>0.5) identified in cochlear pericytes.

| No. | PANTHER pathways | % of gene in the list | | −Log10 (FDR (false discovery rate)) |
| --- | --- | --- | --- | --- |
| | | *Mus musculus* (REF) | Pericyte | |
| 1 | Inflammation mediated by chemokine and cytokine signaling pathway | 1.181979 | 5.072464 | 14.88941 |
| 2 | CCKR signaling map | 0.74101 | 3.804348 | 13.30277 |
| 3 | Angiogenesis | 0.813747 | 3.713768 | 11.61261 |
| 4 | Heterotrimeric G-protein signaling pathway-Gq alpha and Go alpha mediated pathway | 0.554621 | 2.717391 | 8.9914 |
| 5 | Heterotrimeric G-protein signaling pathway-Gi alpha and Gs alpha mediated pathway | 0.722826 | 2.98913 | 8.364516 |
| 6 | Wnt signaling pathway | 1.418375 | 4.257246 | 8.073658 |
| 7 | Gonadotropin-releasing hormone receptor pathway | 1.068327 | 3.623188 | 8.066513 |
| 8 | T cell activation | 0.409147 | 2.173913 | 7.876148 |
| 9 | Apoptosis signaling pathway | 0.563713 | 2.445652 | 7.314258 |
| 10 | Endothelin signaling pathway | 0.377324 | 1.992754 | 7.218245 |
| 11 | B cell activation | 0.322771 | 1.811594 | 6.931814 |
| 12 | VEGF signaling pathway | 0.304587 | 1.630435 | 5.958607 |
| 13 | Oxytocin receptor mediated signaling pathway | 0.272764 | 1.449275 | 5.242604 |
| 14 | Histamine H1 receptor mediated signaling pathway | 0.209119 | 1.268116 | 5.136677 |
| 15 | PI3 kinase pathway | 0.245488 | 1.358696 | 5.135489 |
| 16 | Thyrotropin-releasing hormone receptor signaling pathway | 0.286403 | 1.449275 | 5.086716 |
| 17 | Alzheimer disease-presenilin pathway | 0.577351 | 1.992754 | 4.640165 |
| 18 | Integrin signaling pathway | 0.863754 | 2.445652 | 4.315155 |
| 19 | Alzheimer disease-amyloid secretase pathway | 0.300041 | 1.358696 | 4.308919 |
| 20 | 5HT2 type receptor mediated signaling pathway | 0.309133 | 1.358696 | 4.194499 |
| 21 | Muscarinic acetylcholine receptor 1 and 3 signaling pathway | 0.272764 | 1.268116 | 4.154902 |
| 22 | PDGF signaling pathway | 0.650089 | 1.992754 | 4.017277 |
| 23 | Heterotrimeric G-protein signaling pathway-rod outer segment phototransduction | 0.177297 | 0.996377 | 3.886057 |
| 24 | EGF receptor signaling pathway | 0.618266 | 1.902174 | 3.882729 |
| 25 | Interleukin signaling pathway | 0.395508 | 1.449275 | 3.721246 |
| 26 | Muscarinic acetylcholine receptor 2 and 4 signaling pathway | 0.272764 | 1.177536 | 3.636388 |
| 27 | FGF signaling pathway | 0.554621 | 1.721014 | 3.603801 |

*Table 1 continued on next page*

*Table 1 continued*

| No. | PANTHER pathways | % of gene in the list | | −Log10 (FDR (false discovery rate)) |
|---|---|---|---|---|
| | | *Mus musculus* (REF) | Pericyte | |
| 28 | Cytoskeletal regulation by Rho GTPase | 0.363686 | 1.358696 | 3.595166 |
| 29 | Nicotinic acetylcholine receptor signaling pathway | 0.450061 | 1.449275 | 3.19382 |
| 30 | p53 pathway feedback loops 2 | 0.227304 | 0.996377 | 3.157391 |
| 31 | ATP synthesis | 0.031823 | 0.452899 | 3.123782 |
| 32 | Axon guidance mediated by netrin | 0.159113 | 0.815217 | 3.010105 |
| 33 | Beta1 adrenergic receptor signaling pathway | 0.209119 | 0.905797 | 2.886057 |
| 34 | Enkephalin release | 0.168205 | 0.815217 | 2.882729 |
| 35 | GABA-B receptor II signaling | 0.168205 | 0.815217 | 2.869666 |
| 36 | Metabotropic glutamate receptor group II pathway | 0.213665 | 0.905797 | 2.832683 |
| 37 | Insulin/IGF pathway-protein kinase B signaling cascade | 0.177297 | 0.815217 | 2.772113 |
| 38 | Blood coagulation | 0.236396 | 0.905797 | 2.551294 |
| 39 | FAS signaling pathway | 0.154567 | 0.724638 | 2.522879 |
| 40 | Nicotine pharmacodynamics pathway | 0.159113 | 0.724638 | 2.462181 |
| 41 | Dopamine receptor mediated signaling pathway | 0.25458 | 0.905797 | 2.376751 |
| 42 | Beta2 adrenergic receptor signaling pathway | 0.209119 | 0.815217 | 2.366532 |
| 43 | Alpha adrenergic receptor signaling pathway | 0.10456 | 0.543478 | 2.095284 |
| 44 | Axon guidance mediated by Slit/Robo | 0.113652 | 0.543478 | 1.978811 |
| 45 | TGF-beta signaling pathway | 0.459154 | 1.177536 | 1.978811 |
| 46 | p53 pathway | 0.400055 | 1.086957 | 1.974694 |
| 47 | Androgen/estrogene/progesterone biosynthesis | 0.077283 | 0.452899 | 1.970616 |
| 48 | Endogenous cannabinoid signaling | 0.113652 | 0.543478 | 1.970616 |
| 49 | Cortocotropin releasing factor receptor signaling pathway | 0.154567 | 0.634058 | 1.970616 |
| 50 | Histamine H2 receptor mediated signaling pathway | 0.118198 | 0.543478 | 1.931814 |
| 51 | 5HT1 type receptor mediated signaling pathway | 0.209119 | 0.724638 | 1.910095 |
| 52 | Opioid proopiomelanocortin pathway | 0.168205 | 0.634058 | 1.853872 |
| 53 | Hypoxia response via HIF activation | 0.12729 | 0.543478 | 1.821023 |
| 54 | TCA cycle | 0.054553 | 0.362319 | 1.754487 |
| 55 | Cadherin signaling pathway | 0.736464 | 1.539855 | 1.686133 |
| 56 | Pyruvate metabolism | 0.059099 | 0.362319 | 1.669586 |
| 57 | Triacylglycerol metabolism | 0.009092 | 0.181159 | 1.446117 |

*Table 1 continued on next page*

*Table 1 continued*

| No. | PANTHER pathways | % of gene in the list | | −Log10 (FDR (false discovery rate)) |
|---|---|---|---|---|
| | | *Mus musculus* (REF) | Pericyte | |
| 58 | Opioid prodynorphin pathway | 0.163659 | 0.543478 | 1.407823 |
| 59 | Opioid proenkephalin pathway | 0.172751 | 0.543478 | 1.321482 |

The online version of this article includes the following source data for table 1:

**Source data 1.** Gene list of cochlear pericytes identified by RNA-seq analysis.

RNA, lipids, and metabolites. These are transported to surrounding cells (*Bang and Thum, 2012*) to effect biological function in the receiving cells (*Dai et al., 2020*; *Kalluri and LeBleu, 2020*). To determine whether cochlear pericyte-derived exosomes contribute to the regulation of vascular and neuronal growth in spiral ganglia, we first explored the properties of EVs isolated from pericyte-conditioned culture media. EVs were purified from the collected media by ultrafiltration and size exclusion separation, as illustrated in *Figure 6A*. Using nanoparticle tracking analysis (NTA), a widely employed technique for the characterization of particles in liquids (*Dragovic et al., 2011*), we found a rich population of exosome-sized (~50–150 nm) particles, 96.4% of which are labeled with Exoglow (a membrane EV marker kit), as shown in *Figure 6B*. Further transmission electron microscopy (TEM) images showed a classic cup-shaped structure in these membrane vesicles, which is consistent with previously described exosomes from other cell types, such as HCT116 cells (*Jung and Mun, 2018*) and utricles (*Breglio et al., 2020*). Moreover, proteomic analysis of purified pericyte-derived exosomes identified 580 proteins, 496 of which overlapped with genes that we identified in cochlear pericytes (*Figure 6C*), including the common exosome markers CD9, CD63, CD81, and Tsg101, as shown in *Table 2*. Together, these results indicate the pericytes constitutively release exosomes.

We ran Gene Ontology (GO) enrichment analysis on these proteins using the PANTHER classification system, followed by removal of redundant ontology terms using Reduce and Visualize GO (REVIGO; *Supek et al., 2011*). Analysis by cellular component class identified enrichment of 22 GO terms, including 'extracellular exosomes' (GO: 0070062), neurotransmission related terms such as 'myelin sheath' (GO: 0043209) and 'presynaptic membrane' (GO: 0042734) (*Figure 6D*). Although analysis by PANTHER pathways did not identify significantly enriched pathways, we listed the identified pathways in the order of relative protein number (*Figure 6E*) and found that the angiogenesis related ('Angiogenesis,' P00005; 'VEGF signaling pathway,' P00056) and neurodegeneration related ('Parkinson disease,' P00049; 'Huntington disease,' P00029) pathways were among the top ranks. These pathways are consistent with pathways we identified in cochlear pericytes, as shown in *Figure 5A*. Collectively, these data strongly indicate that cochlear pericyte-derived exosomes participate in regulation of vascular and neuronal growth in the spiral ganglion.

## Cochlear pericyte-derived exosomes promote vascular and neuronal growth via VEGFR2 signaling

VEGF/VEGFR2 signaling plays a key role in formation and growth of blood vessels but is also implicated in neurodegeneration (*Storkebaum and Carmeliet, 2004*). Our bulk RNA-seq analysis identified VEGF-A expression in cochlea pericytes, as did RT-PCR (*Figure 7A*). Although VEGF-C was also in the list, the expression was very low (RPKM = 0.07301). ELISA protein analysis showed 3.696±0.003 ng of VEGF-A/ $8 \times 10^5$ pericytes released into 4 ml of culture medium without growth factors after 3 d culture (*Figure 7B*). In contrast, when we depleted the pericytes in vivo, we found significantly less production of VEGF-A in the cochlea (*Figure 7C*). The immunofluorescence staining further corroborated the VEGFA expression in pericytes in the spiral ganglion region (*Figure 7D*).

Is VEGF-A largely transported by pericyte-derived exosomes or is it directly secreted into the medium? To determine the answer, the exosome and non-exosomal supernatant fraction of the pericyte culture medium were isolated by differential ultracentrifugation and analyzed by western blot (*Figure 7E*). We found that both PDGFRβ (a pericyte cellular membrane marker) and VEGF-A were expressed in the exosomes but not detected in the non-exosomal supernatant fraction (*Figure 7F*). We next asked about the receptor(s) through which VEGF-A exerts its effect on vascular and neuronal

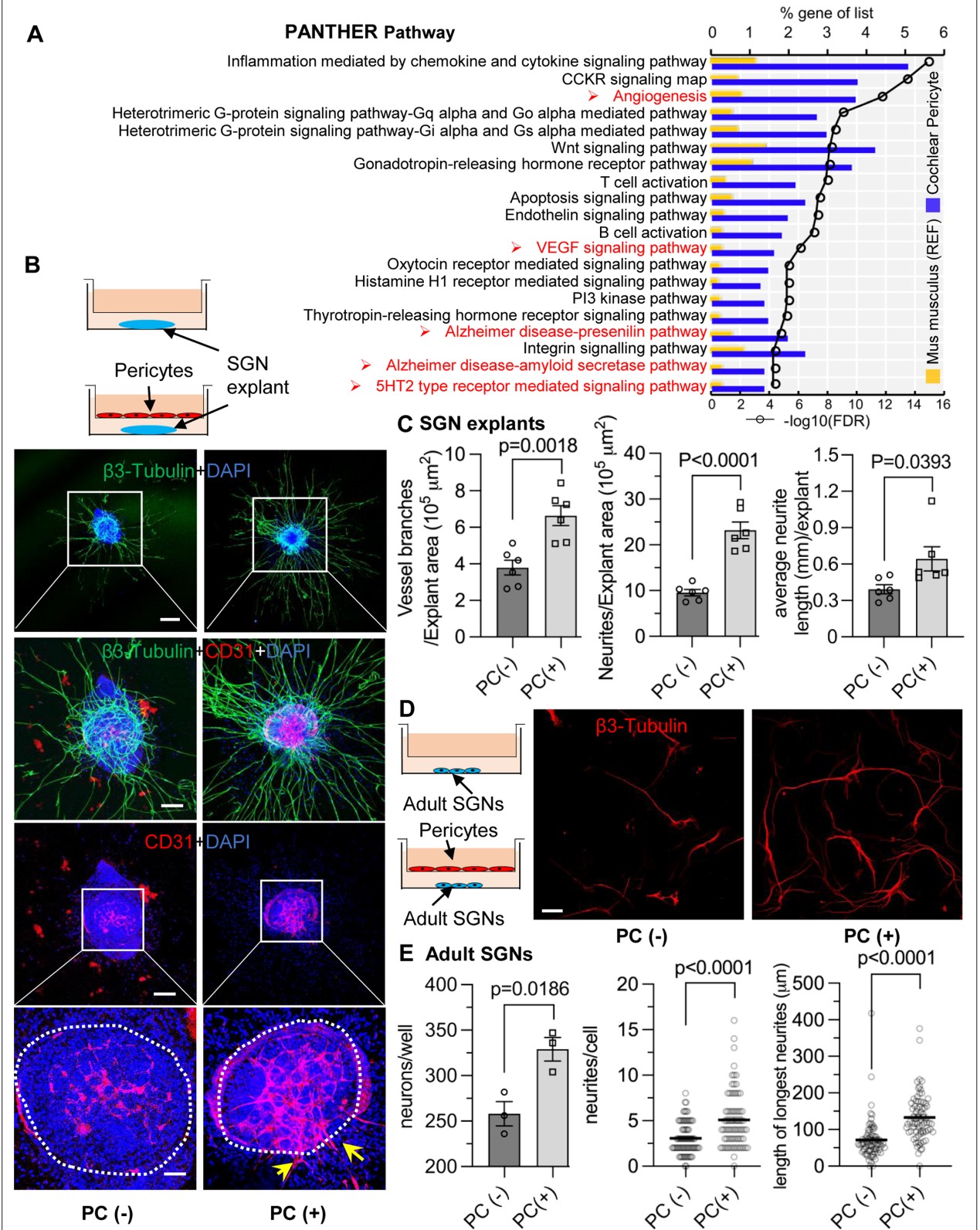

**Figure 5.** Pericytes promote both vascular and neuronal growth in spiral ganglia in vitro. (**A**) The 20 most significant overrepresented PANTHER pathways for genes (RPKM>0.5) identified in cochlear pericytes. (**B**) Neonatal spiral ganglion neuron (SGN) explants co-cultured with pericytes show robust SGN dendritic growth (green, labeled for β-III Tubulin) and new vessel growth (red, labeled for CD31). (**C**) There are significant differences in number of new vessel branches and in number and length of dendritic fibers in the two groups (n=6, $p_{Vascular branch \#/area}$=0.0018, $p_{Neurites \#/area}$<0.0001, $p_{Neurite}$

*Figure 5 continued on next page*

Figure 5 continued

$_{length/explant}$=0.0393, unpaired t test). (**D**) Adult SGNs co-cultured with pericytes show robust SGN dendritic growth (red, labeled for β-III Tubulin). (**E**) There are significant differences in cell survival, and in average neurite number and length, in the two groups (n=3 wells per group, 25 cells per well, $p_{neuron}$ $_{survival}$=0.0186, $p_{Neurites\ \#/cell}$<0.0001, $p_{longest\ neurite\ length/cell}$<0.001, unpaired t test). Data are presented as the mean ± SEM. Scale bars: B, 300 μm (top), 150 μm (middle), 50 μm (bottom); D, 50 μm.

growth. Various neural cells express one or more of the known VEGF receptors and could thus directly respond to VEGF released by neighboring cells (**D'Amore, 2007**; **Ogunshola et al., 2002**; **Teran and Nugent, 2019**). VEGFR2 is the major VEGF receptor with a critical role in angiogenesis (**Abhinand et al., 2016**). It also has been reported to mediate the VEGF signaling in neurons, with roles in neurogenesis, neuronal survival, and axonal growth (**Bellon et al., 2010**; **Luck et al., 2019**). We used immunofluorescence to identify VEGFR2 expression in both SGNs and blood vessels, as shown in **Figure 8A–D**. Our findings lead us to hypothesize that pericyte-derived exosomes promote vascular and neuronal growth in spiral ganglia through a VEGFR2 signaling pathway.

To test our hypothesis, we directly treated neonatal SGN explants and adult SGNs with purified exosomes isolated with total exosome isolation reagent. We found the exosome treatment promoted both vascular and neuronal growth in neonatal SGN explants, which was attenuated by the specific VEGFR2 inhibitor, SU5408 (**Roskoski, 2017**) at 100 nM (n=6, $p_{Vascular\ branch\ \#/area}$<0.0001; $p_{Neurites\ \#/area}$<0.0001; $p_{Neurites\ \#/area}$=0.0268, one-way ANOVA; **Figure 9B and D**). Similarly, exosome treatment also promoted the survival and neurite growth of adult SGNs, which was also arrested by SU5408 (n=4 wells, 25 cells per well, $p_{cell\ survial}$=0.0048, $p_{Neurites\ \#/cell}$<0.0001, $p_{longest\ neurite\ length/cell}$<0.0001, one-way ANOVA; **Figure 9C and E**). The results indicate pericyte-derived exosomes promote vascular and neuronal growth through a VEGFA/VEGFR2 signaling pathway. Dose-response of SU5408 was assessed in a pericyte-SGN explant co-culture model (**Figure 10**).

## Discussion

Pericytes are prevalent on microvessels in the cochlea, but they have not received much research attention. In this investigation, we used an inducible and conditional pericyte depletion mouse model (PDGFRB-CreER$^{T2}$ ROSA26iDTR) to demonstrate that pericytes in the adult ear are critical for the stability of mature vessel beds and viability of SGNs in the peripheral spiral ganglion region. Depletion of pericytes causes loss of vascular volume and SGNs, with concomitant loss of hearing sensitivity. We report on a pericyte-released growth factor, VEGF-A, conveyed by exosomes, shown to strongly mediate and promote SGN survival and growth by binding to VEGFR2 (Flk-1). This study provides the first clear evidence that pericytes have a critical role in vascular and neuronal health in the adult ear. Without the normal disposition of pericytes, animals lose hearing.

Recent research has highlighted the extent to which pericytes play a critical role in vascular function and neuronal protection in the CNS (**Bergers and Song, 2005**; **Brown et al., 2019**). Studies, however, on the role of pericytes in the peripheral nervous system, as in the ear, are limited. In the CNS, loss of pericytes causes brain infarction with diminished capillary perfusion and blood flow (**Armulik et al., 2010**; **Bell et al., 2010**). In the mouse brain, damaged pericytes effectively constrict capillaries and induce a long-lasting reduction in cerebral blood flow, even after recanalization of the larger vessels, and it causes a no-reflow condition (**Hall et al., 2014**; **Kloner et al., 2018**; **Yemisci et al., 2009**). Loss of pericytes also interrupts blood-brain-barrier integrity (**Armulik et al., 2010**; **Bell et al., 2010**; **Nikolakopoulou et al., 2019**). In peripheral organs such as the ear, we previously reported that pericytes control perfusion of cochlear blood flow (**Dai et al., 2010**). Loss of pericytes leads to increased cochlear permeability and reduced endocochlear potential (**Zhang et al., 2021**). In this study, we depleted pericytes in the cochlea in a genetically inducible/conditional adult murine model in which the PDGFRB-CreER$^{T2}$; ROSA26iDTR mice express DTR under pericyte marker *Pdgfrb-Cre* control. The Cre-recombined PDGFR-β$^+$ pericytes expressing DTR are selectively susceptible to ablation on exposure to the DT, while the Cre-negative murine cells, lacking DTR expression, are unaffected. We found depletion of PDGFR-β+ cells reduces pericyte coverage in vascular beds in the spiral ganglion region, where the microvascular network is also densely populated by pericytes (**Jiang et al., 2019**), as shown in **Figure 1**. The depletion has a direct effect on vascular density (**Figure 2**). The results obtained in this study further demonstrate the important role of pericytes in maintaining vascular

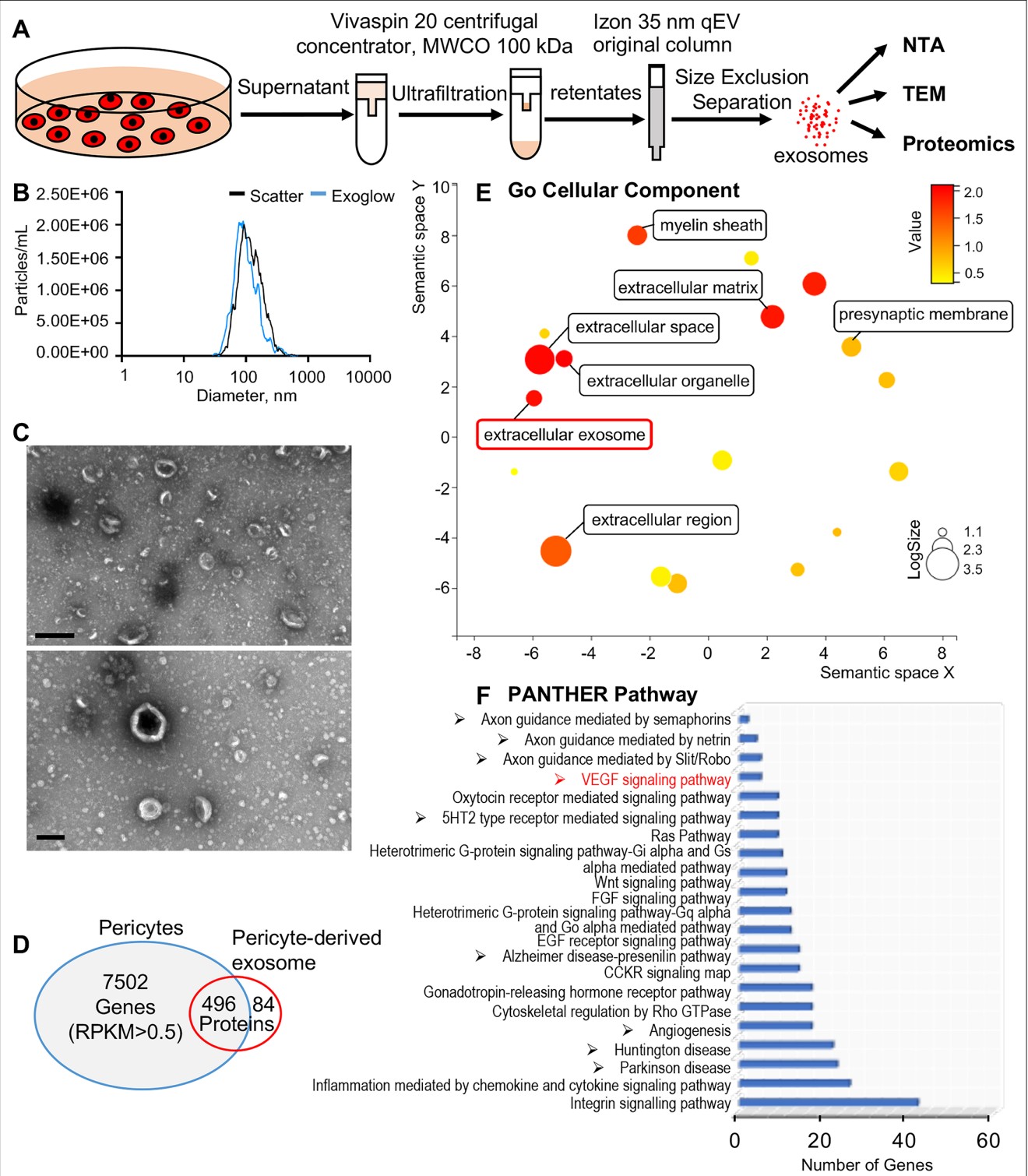

**Figure 6.** Identification of pericyte-derived exosomes. (**A**) Schematic of the exosome purification procedure used to isolate exosomes from pericyte-conditioned culture medium for nanoparticle tracking analysis (NTA), transmission electron microscopy (TEM), and proteomics analysis. Exosomes were isolated via ultracentrifugation followed by size-exclusion separation. (**B**) NTA shows a rich population of exosome-sized (~50–150 nm) particles, 96.4% of which are labeled with Exoglow (a membrane extracellular vesicle [EV] marker kit). (**C**) TEM images show a classic cup-shaped structure membrane vesicle with diameter around 100 nm. (**D**) Proteomics analysis of exosomes identified a total of 570 unique protein families in the exosomes, 496 of which overlap with genes identified in pericyte isolated total RNA. (**E**) The 22 most significantly enriched Gene Ontology (GO) cellular component terms for

*Figure 6 continued on next page*

*Figure 6 continued*

proteins identified in the pericyte-derived exosomes. (**F**) The relative PANTHER pathways identified in pericyte-derived exosome proteins. Scale bars: C, 500 nm (top) and 200 nm (bottom).

The online version of this article includes the following source data for figure 6:

**Source data 1.** Protein families of cochlear pericyte-derived exosomes identified by proteomics analysis.

stability in peripheral systems such as auditory SGNs in the cochlea. However, PDGFR-β is a general membrane protein not only expressed by all cochlear pericytes (*Canis and Bertlich, 2019*; *Shi et al., 2008*) but also by pericytes in other organs. Thus, our PDGFR-β depletion model should deplete all pericytes in different organs and in different regions of the cochlea, including in the region of the blood-labyrinth barrier in the stria vascularis. This causes blood-barrier dysfunction of the adult mouse cochlea, leading to increased cochlear permeability, reduced endocochlear potential, and consequent hearing dysfunction, as we reported in a previous study (*Zhang et al., 2021*). However, with these caveats taken into consideration, pericytes are also well-known to be heterogeneous, organ-oriented, and tissue-specific (*Dias Moura Prazeres et al., 2017*; *Hirschi and D'Amore, 1996*). As one caveat and consistent with a study from *Park et al., 2017*, we have not observed obvious abnormalities of retina blood vessels in our adult pericyte depletion model (*Appendix 1—figure 2*). The different results obtained from pericyte depletion does, however, support the notion that pericytes display considerable phenotypic and functional heterogeneity in different tissues and organs (*Sims, 2000*).

SGNs are neural elements transmitting localized electrical activity from HCs to second-order neurons in the cochlear nucleus (*Coate et al., 2019*). By labeling SGNs with β-III tubulin, a specific marker for neuronal cytoskeleton is commonly used to distinguish neurons from other cell types (*Katsetos et al., 2003*), we found that loss of pericytes affects SGN viability and causes significant loss of SGNs across all turns. Pericyte loss also significantly decreases expression of β-III tubulin protein in SGNs in pericyte-depleted animals, particularly at the middle and basal turns, as shown in *Figure 4*. Tubulin is the primary structural protein in microtubules. In fact, the tubulin assembly is essential to the structure, stability, and function of the neurons. A gradual loss of neuronal microtubule mass is seen in many neuro-degenerative diseases (*Baas et al., 2016*). In ABR functional studies, delay in latency and low amplitude is first seen at the basal turn and moves to regions corresponding to middle and low frequencies, as shown in *Figure 3*. The mechanism underlying the vulnerability of SGNs at the basal turn is not known. However, the significantly decreased expression of β-III tubulin protein in SGNs at the basal and middle turns may indicate that SGN damage is more severe than at the apical turn. In addition, a recent single-cell analysis study of mouse SGNs showed there were three functionally distinct subtypes of SGN, each displaying molecular variation along the tonotopic axis (*Shrestha et al., 2018*). Sensitivity at the different turns to pericyte loss could also be due to SGN diversity and the variance in SGN vulnerability to environmental perturbation. Further study is needed to define the mechanisms of SGN vulnerability.

What is the underlying mechanism of pericyte-loss-related damage to SGNs? Our data suggest both blood flow dysfunction and direct disruption of pericyte-SGN communication are at issue. It is well-documented that pericyte loss triggers primary vascular dysfunction, leading to neurodegeneration (*Winkler et al., 2011*). For example, in the CNS, pericyte loss induces vascular leakage, insufficient blood perfusion, and tissue hypoxia, ultimately leading to neuron degeneration (*Quaegebeur et al., 2010*; *Sagare et al., 2013*; *Winkler et al., 2011*; *Zlokovic, 2011*). SGNs are vulnerable to interruption of the blood supply since the energy demand of SGNs is directly met by the surrounding

**Table 2.** Common protein markers for exosomes were identified in the pericyte-derived exosomes.

| Protein | Gene names | # PSMs | Log2 LFQ (label-free quantitation) intensity |
|---|---|---|---|
| CD81 antigen | *Cd81* | 26 | 29.6153 |
| CD63 antigen | *Cd63* | 3 | 26.1908 |
| CD9 antigen | *Cd9* | 16 | 29.2576 |
| Tumor susceptibility gene 101 | *Tsg101* | 5 | 25.7598 |

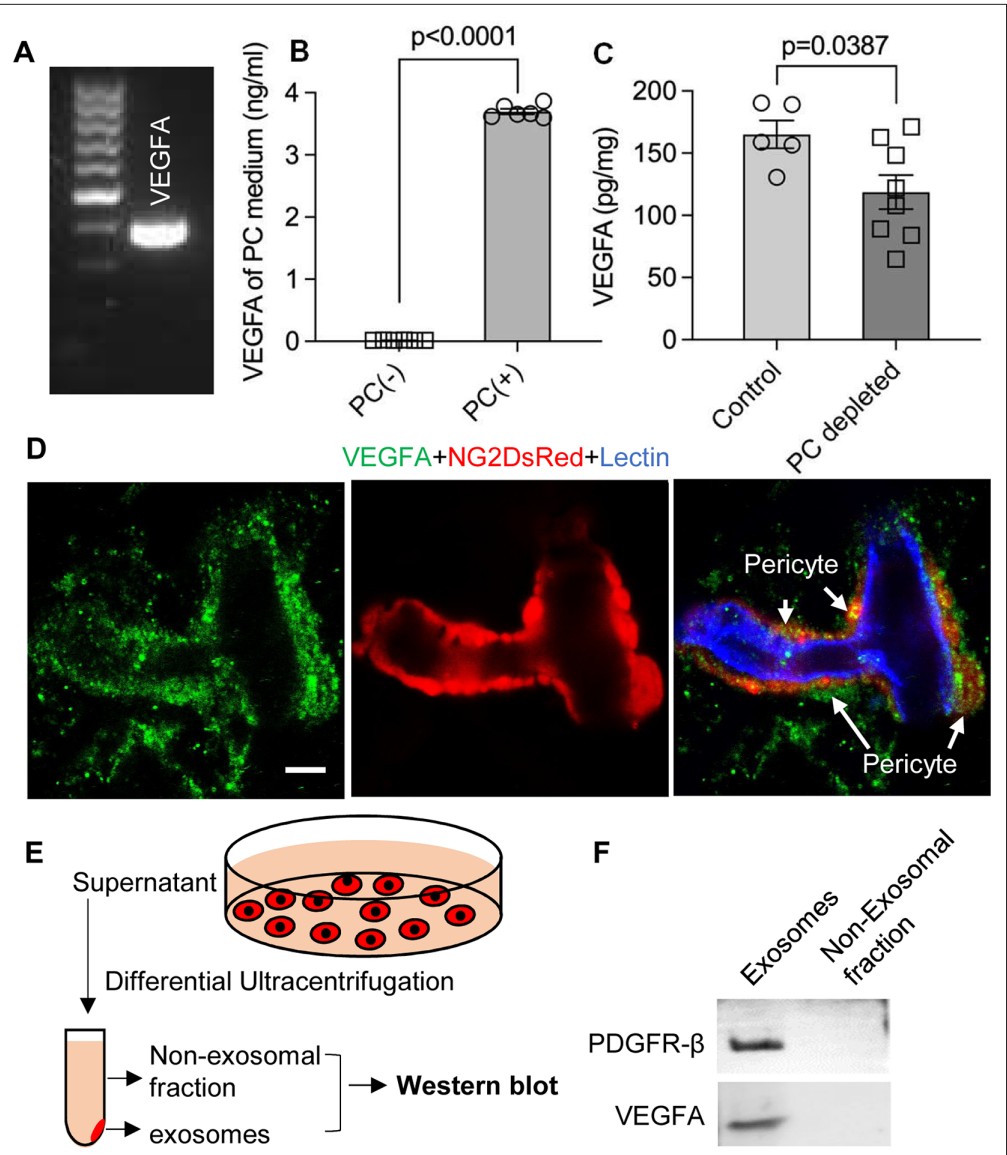

**Figure 7.** Pericytes release VEGFA through exosomes. (**A**) mRNA expression of VEGF-A in primary cochlear pericytes. (**B**) VEGF-A production assessed by ELISA at day 3 in the control and pericyte containing culture medium (n=6, p<0.0001, unpaired t test). (**C**) VEGFA expression level assessed by ELISA in the cochlea of control and pericyte-depleted mice ($n_{control}$ = 5, $n_{pericyte depletion}$=8, p=0.0387, unpaired t test). (**D**) Representative confocal images showing VEGF-A expression in pericytes. (**E**) Exosomes and the non-exosomal fraction (supernatant) were purified from pericyte-conditioned media using differential ultracentrifugation for western blot analysis. (**F**) Western blot showing the expression of VEGFA in exosomes. Similarly, PDGFR-β (pericyte membrane marker) was detected exclusively in exosomes. Data are presented as the mean ± SEM. Scale bars: D, 5 μm.

The online version of this article includes the following source data for figure 7:

**Source data 1.** Original uncropped blots of VEGFA and PDGFR-β, and raw gel data of whole protein staining with the relevant bands clearly labeled.

**Source data 2.** Original uncropped blots of VEGFA and PDGFR-β, and raw gel data of whole protein staining with the relevant bands clearly labeled.

blood circulation (*Jiang et al., 2019*; *Mei et al., 2020*). In this study, our data demonstrate reduction in vascular density and change in vessel diameter in an adult PDGFRβ+ pericyte depletion model. Our data is supported by Kramann et al., where it was found pericyte loss induces capillary rarefaction in acute kidney injury (*Guzmán-Hernández et al., 2014*). Detachment of pericytes from endothelial

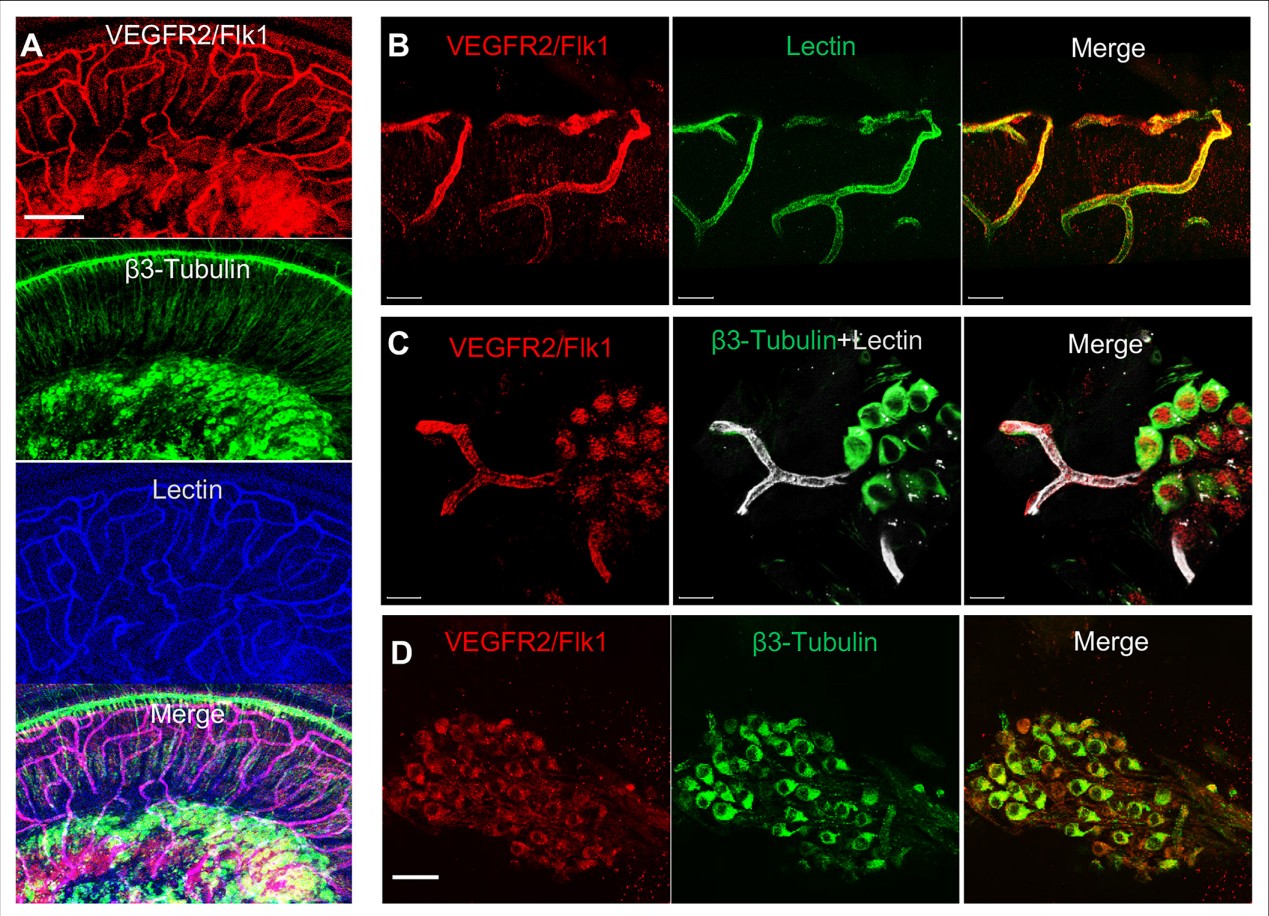

**Figure 8.** VEGFR2 expression in the spiral ganglion region. (**A–C**) Representative confocal images of a cochlear whole mount under low (**A**) and high (**B and C**) magnification. (**D**) Cross section showing VEGFR2 (red) is positively expressed in both spiral ganglion neurons (SGNs; labeled for β-III Tubulin) and blood vessels (labeled for lectin). Scale bars: E, 100 µm; F, 30 µm; G, 20 µm; H, 50 µm.

cells can interrupt cross-talk signaling and destabilize capillaries, as we earlier reported (*Hou et al., 2020*). Without the presence of pericytes, tight junction adhesive proteins such as ZO-1 are significantly reduced (*Neng et al., 2013b*). The increased vascular permeability and loss of vascular volume cause cochlear tissue edema and hypoxia, damaging neurons (SGNs) and sensory HCs, as we found in both this study and a previous study of sensory HCs (*Zhang et al., 2021*), although we did not measure vascular permeability in this study. In addition to the effect of blood flow on SGNs, we previously demonstrated active communication between pericytes, endothelial cells, and SGNs in the cochlea (*Jiang et al., 2019*), also shown in *Figure 1*. Pericytes release many growth factors relevant to maintenance of organ homeostasis (*Gaceb et al., 2018a*; *Gaceb et al., 2018b*; *Gaceb and Paul, 2018c*). In this study, we used bulk-RNA-seq analysis on the purified cochlear pericytes to identify the different vascular- and neuronal-growth factors in our mouse cochlear pericyte dataset. We found pericyte-related angiogenesis and neuroprotective pathways overrepresented (*Figure 5A*). In pericyte co-culture models of neonatal SGN explants and adult SGNs, we found pericytes to induce robust vascular and neuronal dendritic growth in the neonatal SGN explants and to increase cell survival and neurite growth in the adult SGNs, as shown in *Figure 5B–E*. The results suggest direct intercellular communication between pericytes and SGNs in addition to direct effects exerted by pericytes on vascular function.

How do pericytes communicate with endothelial cells and SGNs? Many cells release exosomes, nano-sized EVs (50–150 nm diameter), which transport cargo such as nucleic acids, proteins, and lipids. These exosomes have significant physiological effects in the recipient cells (*Bang and Thum, 2012*; *Colombo et al., 2014*; *Zhang et al., 2019*). Pericyte-released exosomes were first reported by *Gaceb and Paul, 2018c*. Pericyte-derived exosome physiology, including characterization of size and

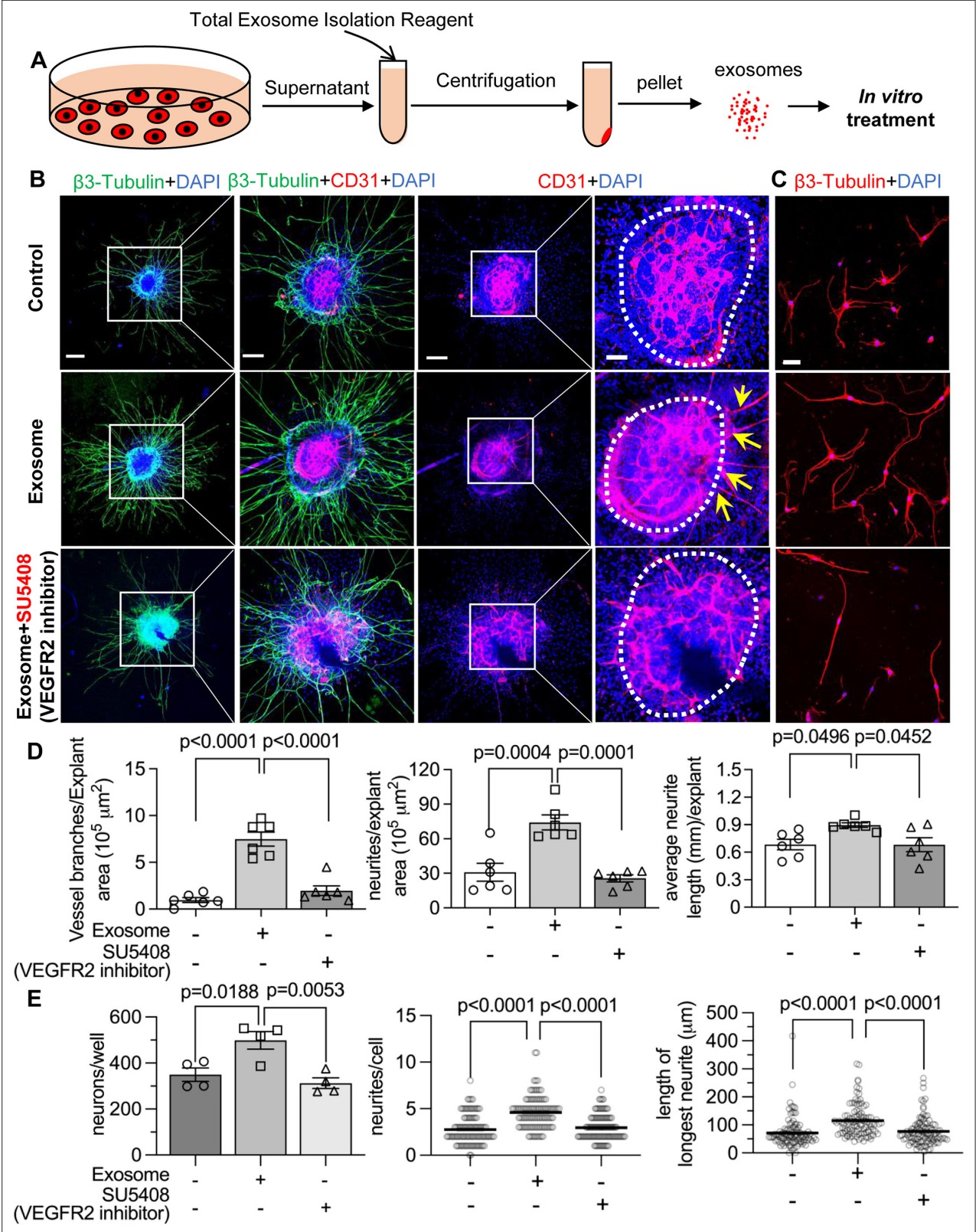

**Figure 9.** Pericyte-derived exosomes promote angiogenesis and spiral ganglion neuron (SGN) growth through VEGFR2 signaling. (**A**) Exosomes were purified from pericyte-conditioned media using total exosome isolation reagent for in vitro treatment. (**B**) Compared to the control group, exosome treated SGN explants showed robust SGN dendritic growth (green, labeled for β-III Tubulin) and new vessel growth (red, labeled for CD31). In contrast, both neurogenic and angiogenetic activity were decreased when a VEGFR2 inhibitor, SU5408, was presented in the medium. (**C**) Exosome treated adult

*Figure 9 continued on next page*

*Figure 9 continued*

SGNs showed more SGN dendritic growth (red, labeled for β-III Tubulin) compared to control and VEGFR2 blocked groups. (**D**) There are significant differences in new vessel branch number and in dendritic fiber number and length in the three groups (n=6, p=0.0017). (**E**) There are significant differences in cell survival and in average neurite number and length, in the three groups (n=4 wells per group, 25 cells per well, p=0.0048). One-way ANOVA followed by Tukey's multiple comparison test, individual p values of different group comparisons are labeled on the graph. Data are presented as the mean ± SEM. Scale bars: B, 300 μm (left), 150 μm (center), 50 μm (right); C, 50 μm.

morphology, remains limited. However recent studies have amply demonstrated that pericyte-derived exosomes participate in regulation of microvascular function under different pathological conditions (*Ye et al., 2021*; *Yuan et al., 2019*). In our study, we observed NG2-positive particles (likely released by pericytes) in the soma of SGNs (*Figure 1E*). The pericyte-derived particles displayed the classic cup-shaped morphology (*Figure 6B and C*). Further proteomic analysis of these particles identified 580 proteins enriched in the GO of cellular components in extracellular exosomes, including proteins relating to angiogenesis, neurodegeneration, and neuroprotection (*Figure 6E–F*). Our findings are strong indication of participation of cochlear pericyte-derived exosomes in intercellular communication with spiral ganglion.

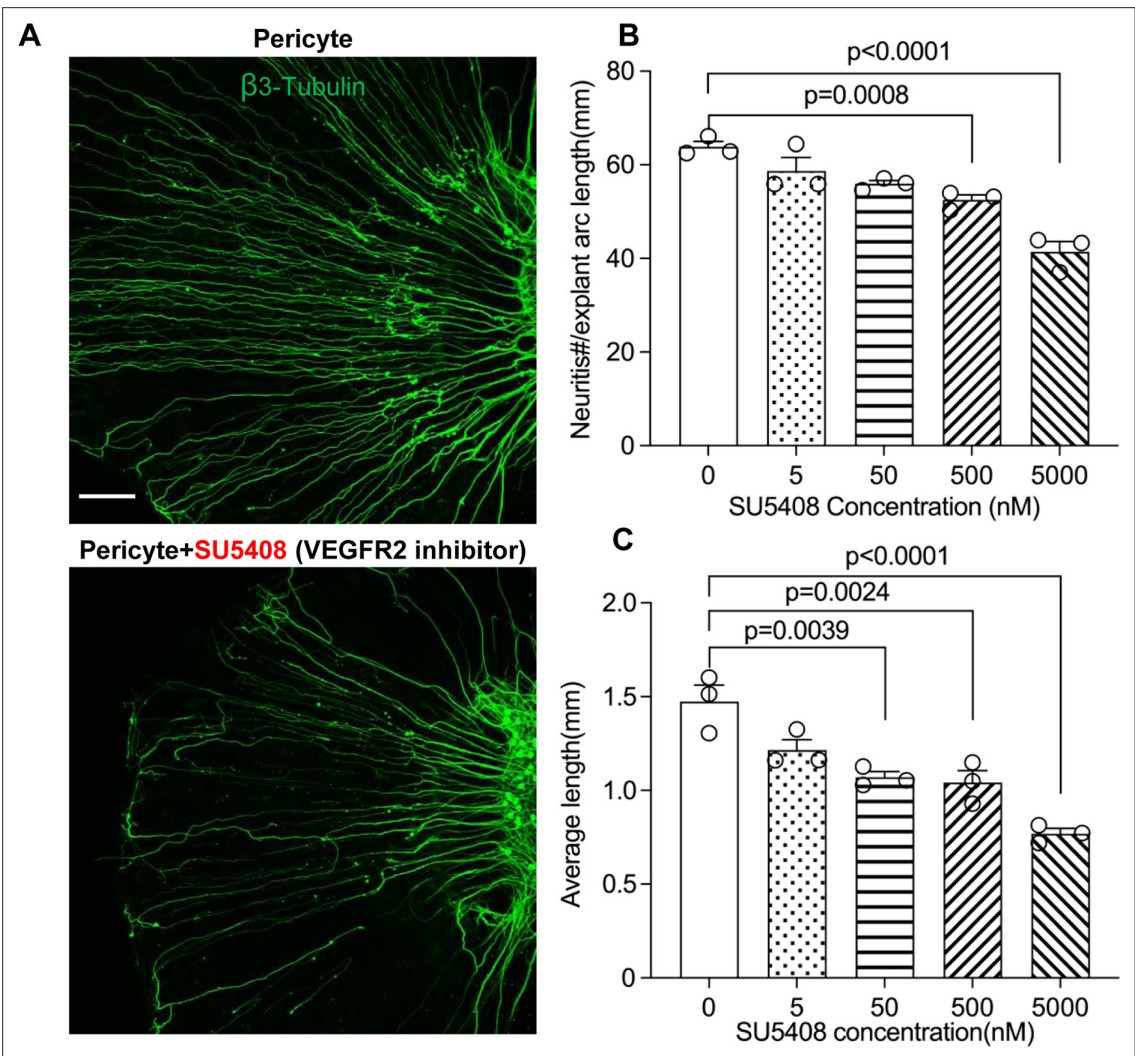

**Figure 10.** Neuronal dendritic growth is reduced when VEGFR2 is blocked in a pericyte-spiral ganglion neuron (SGN) explant co-culture model. (**A**) Representative images showing the pattern of SGN dendritic growth under different experimental conditions. (**B and C**) Number and length of dendritic fibers when the VEGFR2 receptor is blocked. A dose-dependent relationship is shown (n=3, $p_{number}<0.0001$, $p_{length}<0.0001$, one-way ANOVA followed by a Tukey's multiple comparison test, individual p values between different groups labeled on the graph). Data are presented as the mean ± SEM. Scale bar: A, 150 μm.

How do pericyte-derived exosomes affect angiogenesis and neurogenesis? VEGF/VEGFR2 signaling is a well-defined, classical pathway that plays a critical role in both angiogenesis and neuro-protection (**Storkebaum and Carmeliet, 2004**). VEGFR2 is typically expressed in both vascular cells and SGNs (**Figure 8A–D**). In this study, we used in vitro neonatal SGN explant and adult SGN tissue culture models to specifically investigate the VEGF-A controlled angiogenesis signaling pathway in relation to vascular and neuronal growth. Our results show blockage of the VEGFR2 receptor with a specific VEGFR2 inhibitor, SU5408 (**Roskoski, 2017**), dramatically halts angiogenesis and neuronal growth (**Figure 9**), indicating that pericyte-derived VEGF-A-containing exosomes promote both vascular and neuronal growth through a VGFR2 signaling pathway. However, the exact mechanism of the exosome-mediated VEGF-A/VEGFR2 signaling pathway remains unclear. For example, studies have shown the shedding of VEGFA-165 from the cell surface (together with other membrane components) appears to be a unique mechanism by which some VEGF is delivered to the surroundings to exert its known biological actions (**Guzmán-Hernández et al., 2014**). The interaction of isoform VEGF189 with the EV surface membrane profoundly increased the ligand half-life and stability (**Ko et al., 2019**). Although we have determined VEGF-A is predominant in the exosomes of cultured PC media, suggesting the VEGF-A is delivered by exosomes, it still needs further investigation to better understand the difference between the pericyte-released VEGFA in diffusible isoform and exosomes. It is also unclear how VEGF-A contained in exosomes binds to Flk1 receptors in recipient cells. Studies demonstrate EVs interface with neighboring cells through different modes, including direct receptor-binding without internalization of EVs, phagocytosis, macropinocytosis, internalization by clathrin-caveolae- and lipid raft-mediated endocytosis (**Mulcahy et al., 2014**). Once internalized, the EVs release their cargo in the cytoplasm, and the cell-internalized cargo then regulates the cell at the transcription or translation level (**Abels and Breakefield, 2016**; **Berumen Sánchez et al., 2021**). It has also been reported that VEGF presents on the surface of EVs through binding to surface proteins such as heparan and heat shock protein 90 (HSP90), which can directly stimulate the cellular domain of Flk1 and induce phosphorylation (**Ko et al., 2019**; **Ko and Naora, 2020**). Overall, however, the data clearly indicate cochlear angiogenesis and neurogenesis require VEGF-A signaling. A recent study by Okabe et al. reports that neuron-derived VEGF contributes to cortical and hippocampal development but is independent of VEGFR1/2-mediated neurotropism (**Okabe et al., 2020**). They conclude that neuron-derived VEGF contributes to cortical and hippocampal development likely through angiogenesis independent of any direct neurotrophic effects mediated by VEGFR1 and 2. Although there is still no consensus on whether cochlear angiogenesis affects auditory peripheral neurogenesis, our co-culture model of pericyte-SGN explants and adult SGNs does provide evidence the VEGF-A/VEGFR2 signal strongly promotes auditory SGN dendritic growth, as shown in **Figure 9**. In addition, cochlear pericyte-derived exosomes contain numerous other proteins, as we identified in our proteomic analysis. For example, they contain messenger RNA and micro-RNAs not investigated in this study. Some of these may also be playing an important role in the regulation of angiogenesis, SGN development, and neurite growth. However, it must be emphasized that treatment with the VEGFR2 inhibitor almost completely blocked the effect of the exosomes relative to control groups (see **Figure 9**). This suggests the VEGF-A/VEGFR2 signaling is the dominant factor regulating vascular and neuronal growth. However, this does not obviate the possibility the underlying mechanism of pericyte-related neuroprotection is more complicated than what we have discovered in this study. A future in vivo study in a conditional SGN-specific VEGFR2 knockdown animal model will give more conclusive results.

In summary, our data provide the first clear-cut experimental evidence that pericytes are essential for the maintenance of cochlear vascular stability and SGN viability in adults. Loss of pericytes disrupts vascular structure and damages SGNs. The damage is related to the impairment of VEGF-A mediated communication between pericytes, endothelial cells, and SGNs, as illustrated in **Figure 11**. These findings demonstrate for the first time that pericyte-released exosomes have a major role in vascular and neuronal health.

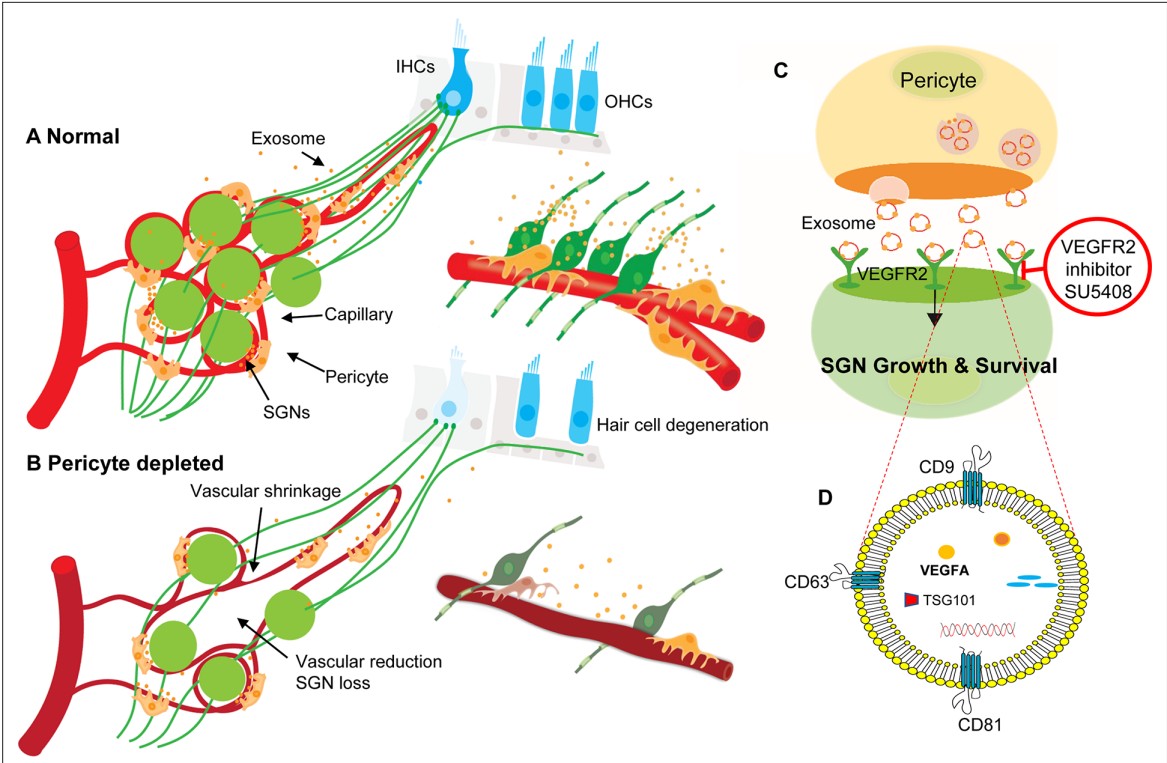

**Figure 11.** Hypothesized mechanisms of pericyte-mediated effects on spiral ganglion neuron (SGN) growth and survival. (**A**) Under normal conditions, pericytes participate in the maintenance of SGN health through two parallel pathways: (1) One maintains vascular stability and function and (2) the other 'nourishes' SGNs through release of exosomes. (**B**) Pericyte depletion causes reduction in vascular volume as well as dysfunction, including loss of SGNs and hair cells (*Buch et al., 2005*), and, in consequence, hearing loss. (**C**) VEGF-A-carrying exosomes interact with VEGFR2 on the SGNs to stimulate growth and promote survival, which can be arrested by the specific VEGFR2 inhibitor, SU5408. (**D**) The schematic model shows the molecular structure of exosomes and includes mention of common exosome markers such as CD81, CD63, CD9, and Tsg101 and exosome cargo such as proteins, DNAs, RNAs, lipids, and metabolites.

# Materials and methods

## Key resources table

| Reagent type (species) or resource | Designation | Source or reference | Identifiers | Additional information |
|---|---|---|---|---|
| Strain, strain background (*Mus musculus*) | C57BL/6J | The Jackson Laboratory | RRID:IMSR_JAX:000664 | |
| Strain, strain background (*M. musculus*) | B6.Cg-Gt(ROSA)26Sor^{tm9(CAG-tdTomato)Hze}/J | The Jackson Laboratory | RRID:IMSR_JAX:007909 | |
| Strain, strain background (*M. musculus*) | B6.Cg-Tg(Pdgfrb-cre/ERT2)6096Rha/J | The Jackson Laboratory | RRID:IMSR_JAX:029684 | |
| Strain, strain background (*M. musculus*) | C57BL/6-Gt(ROSA)26Sor^{tm1(HBEGF)Awai}/J | The Jackson Laboratory | RRID:IMSR_JAX:007900 | |
| Strain, strain background (*M. musculus*) | Tg(Cspg4-DsRed.T1)1Akik/J | The Jackson Laboratory | RRID:IMSR_JAX:008241 | |
| Antibody | anti-Desmin [Y66] (Rabbit monoclonal) | Abcam | Cat#: ab32362 | 1:50 |

*Continued on next page*

*Continued*

| Reagent type (species) or resource | Designation | Source or reference | Identifiers | Additional information |
|---|---|---|---|---|
| Antibody | anti-β-III Tubulin [EP1569Y] (Rabbit monoclonal) | Abcam | Cat#: ab52623 | 1:200 |
| Antibody | anti-CD31 [MEC 7.46] (Rat monoclonal) | Abcam | Cat#: ab7388 | 1:100 |
| Antibody | anti-VEGFR2 [EPR21884-236] (Rabbit monoclonal) | Abcam | Cat#: ab233693 | 1:500 |
| Antibody | anti-VEGFA (Rabbit polyclonal) | Abcam | Cat#: ab51745 | 1:400 |
| Antibody | anti-PDGFRβ [Y92] (Rabbit monoclonal) | Abcam | Cat#: ab32570 | 1:1000-1:5000 |
| Sequence-based reagent | B6.Cg-Tg(Pdgfrb-cre/ERT2)6096Rha/J F | The Jackson Laboratory | PCR primers | GAA CTG TCA CCG GGA GGA |
| Sequence-based reagent | B6.Cg-Tg(Pdgfrb-cre/ERT2)6096Rha/J R | The Jackson Laboratory | PCR primers | AGG CAA ATT TTG GTG TAC GG |
| Sequence-based reagent | B6.Cg-Tg(Pdgfrb-cre/ERT2)6096Rha/J internal positive control F | The Jackson Laboratory | PCR primers | CAA ATG TTG CTT GTC TGG TG |
| Sequence-based reagent | B6.Cg-Tg(Pdgfrb-cre/ERT2)6096Rha/J internal positive control R | The Jackson Laboratory | PCR primers | GTC AGT CGA GTG CAC AGT TT |
| Sequence-based reagent | Mouse VEGFA F | IDT | PCR primers | GCAGCGACAAGGCAGACTA |
| Sequence-based reagent | Mouse VEGFA R | IDT | PCR primers | GGTCCGATGCAAGATCCCAA |
| Peptide, recombinant protein | Diphtheria toxin | Sigma | Cat#: D0564 | 10 ng/g body weight |
| Peptide, recombinant protein | Dispase II | Sigma | Cat#: D4693 | |
| Peptide, recombinant protein | Collagenase I | ThermoFisher | Cat#: 17018029 | |
| Peptide, recombinant protein | DNase I | Sigma | Cat#: 10104159001 | |
| Peptide, recombinant protein | 100× Penicillin-Streptomycin Solution | Invitrogen/Gibco | Cat#: 15140–122 | |
| Commercial assay or kit | Invitrogen Total Exosome Isolation Reagent (from cell culture media) | ThermoFisher | Cat#: 4478359 | |
| Commercial assay or kit | SuperSignal West Femto Duration Substrate | Thermo Fisher Scientific | Cat#: A38554 | |
| Commercial assay or kit | Clontech SMARTer cDNA kit | Clontech Laboratories | Cat#: 634925 | |
| Commercial assay or kit | NEBNext reagents | New England Biolabs | Cat#: E6040 | |
| Commercial assay or kit | RNeasy micro kit | Qiagen | Cat#: 74004 | |
| Commercial assay or kit | SuperScript IV First-Strand Synthesis kit | ThermoFisher | Cat#: 18091050 | |
| Commercial assay or kit | VEGFA ELISA Kit | Abcam | Cat#: ab119565 | |

*Continued on next page*

*Continued*

| Reagent type (species) or resource | Designation | Source or reference | Identifiers | Additional information |
|---|---|---|---|---|
| Commercial assay or kit | BCA protein assay kit | Abcam | Cat#: ab102536 | |
| Commercial assay or kit | ExoGlow-Protein EV Labeling Kit (Green) | SBI | Cat#: EXOGP300A-1 | |
| Chemical compound and drug | Tamoxifen | Sigma | Cat#: T5648 | 75 mg/kg body weight |
| Chemical compound and drug | SU5408 | Abcam | Cat#: ab145888 | |
| Software and algorithm | Sample Size Calculator | N/A | https://clincalc.com/stats/samplesize.aspx | |
| Software and algorithm | ImageJ | NIH | https://imagej.nih.gov/ij/ | |
| Software and algorithm | PANTHER classification system | N/A | http://www.pantherdb.org/ | |
| Software and algorithm | REVIGO | Ruđer Bošković Institute | http://revigo.irb.hr/ | |
| Other | Lectin-DyLight 488 | Vector Laboratories | Cat#: DL-1174 | 20 µg/ml |
| Other | Lectin-DyLight 649 | Vector Laboratories | Cat#: DL-1178 | 20 µg/ml |
| Other | Decal Stat Decalcifier | StatLab | Cat#: 1212–32 | Use directly (contains Hydrogen Chloride, Acid mists, strong inorganic) |
| Other | Antifade Mounting Medium with DAPI | Vector Laboratories | Cat#: H-1200 | Use directly (contains 1 µg/ml of DAPI) |

## Animals

All strains of mice used in this study were originally purchased from Jackson Laboratory, including C57BL/6J (wild type, strain # 000664), B6.Cg-*Gt(ROSA)26Sor^{tm9(CAG-tdTomato)Hze}*/J (ROSA26tdTomato, strain # 007909), B6.Cg-Tg(Pdgfrb-cre/ERT2)6096Rha/J (PDGFRB-CreER^{T2}, strain # 029684), C57BL/6-*Gt(ROSA)26Sor^{tm1(HBEGF)Awai}*/J (ROSA26iDTR, strain # 007900), and Tg(Cspg4-DsRed. T1)1Akik/J (NG2DsRedBAC, Strain # 008241). To verify location of the Pdgfrb/Cre, tdTomato fluorescence reporter mice were created by breeding PDGFRB-CreER^{T2} mice with ROSA26tdTomato mice. CreERT2-mediated recombination was initiated by intraperitoneal injection of TAM at 75 mg/body weight every 24 hr for three consecutive days. The inducible pericyte depletion mouse model (PDGFRB-CreER^{T2+/−}; ROSA26iDTR^{+/−}) was created by crossing PDGFRB-CreER^{T2+/−} transgenic mice with ROSA26iDTR^{+/+} mice. To deplete pericytes, these PDGFRB-CreER^{T2+/−}; ROSA26iDTR^{+/−} mice were given DT intraperitoneally once every 24 hr at a dose of 10 ng/g on four consecutive days after administering the TAM (as illustrated in *Figure 2A and B*). PDGFRB-CreER^{T2−/−}; ROSA26iDTR^{+/−} mice from same litters of PDGFRB-CreER^{T2+/−}; ROSA26iDTR^{+/−} mice were treated with TAM and DT as a control group. All transgenic mice were maintained in the lab, validated, and genotyped for the study. Both male and female mice were used, and all mice used were adults aged between 4~8 wk or postnatal P1~P3. All animal experiments reported were approved by the Oregon Health and Science University Institutional Animal Care and Use Committee (IACUC IP00000968).

## Immunofluorescence of cochlea whole mount

Cochlea whole mount was dissected and stained with fluorescently tagged antibodies as described by *Montgomery and Cox, 2016*. Briefly, cochleae were harvested and fixed in 4% paraformaldehyde (PFA) overnight. After decalcification in Decal Stat Decalcifier overnight at 4°C, each cochlea was carefully dissected into three whole mount turns and incubated in blocking/permeabilization solution containing 0.25% Triton X-100, 10% goat serum (GS) in 1× PBS for 1 hr at room temperature (RT), then transferred into primary antibody solution (diluted in blocking/permeabilization solution) and incubated overnight at 4°C. After three washings with 1× PBS, samples were incubated with the fluorescence-conjugated secondary antibody in the blocking/permeabilization solution for 1 hr at RT,

washed again with 1× PBS three times, and mounted in Antifade Mounting Medium with DAPI on slides.

## Assessment of pericyte coverage and vascular density in the spiral ganglion region

Mice were anesthetized and administered the fluorescent dye Lectin-DyLight 649 diluted in 0.1 M PBS buffer to a concentration of 20 µg/ml (vol. 100 µl) via intravenous retro-orbital sinus for 10 min before they were sacrificed. The cochleae whole mounts were stained with desmin 1:50 and visualized under an FV1000 Olympus confocal microscope with a 10× objective. The pericyte coverage and blood vessel area were calculated using Fiji (ImageJ, NIH, 1.51t) software. Pericyte distribution was defined as pericyte coverage = $\frac{area\ of\ desmin\ (pericyte\ filaments)}{area\ of\ spiral\ lamina}$ . (**Note:** Pericyte intermediate filaments were assessed to obtain pericyte coverage. We did not directly count pericyte number since all pericyte markers, including antibodies for NG2 and PDGFRβ, failed. Adult spiral limbus is very thick. For visualization of the whole mounts, including the vascular networks in the tissue, the tissues would need to be decalcified. However the decalcification process causes deterioration of membrane proteins.) Vascular density was analyzed as previously described (*Jiang et al., 2019*), defined as vascular density = $\frac{area\ of\ blood\ vessels}{area\ of\ spiral\ lamina}$.

## Demonstration of VEGFR2 expression and pericyte distribution in the spiral ganglion region

VEGFR2 expression in the region of the spiral ganglion was demonstrated in wild type mice and pericyte distribution in NG2DsRedBAC mice. Mice were anesthetized and placed on a water heating pad, and the fluorescent dye Lectin-DyLight 488/649 diluted in 0.1 M PBS buffer to a concentration of 20 µg/ml (vol. 100 µl) was administered via intravenous retro-orbital sinus to the animals 10 min before they were sacrificed. The isolated cochleae were fixed in 4% PFA overnight. The next day, tissue samples from the apical turn were isolated and the SGNs exposed after gently breaking the covering bone and removing the chips surrounding the ganglion. The tissue samples were stained with anti-VEGFR2 antibody (1:500) and/or anti-β-III tubulin (1:200) antibody and imaged on an FV1000 Olympus laser-scanning confocal microscope. This 'gentle-bone-breaking' (apex) preparation enables us to preserve the membrane protein affinity for antibodies and capture the fluorescence signal from the DsRed-tagged NG2+ pericytes.

## ABR test

ABR audiometry to pure tones was used to evaluate hearing function before and after DT treatment as previously described (*Zhang et al., 2021*). For measurement of hearing threshold, latency, and amplitude by ABR, 10 ears from 5 animals in both the control and pericyte depleted group were measured by an investigator blind to treatment status. ABR response to a 1 ms rise-time tone burst (30 rep rate/s) at 8, 16, 24, and 32 kHz was recorded. Stimulus SPLs were incremented in 5 dB steps from below threshold to 100 dB SPL. The latency at 20–100 dB and peak-to-peak (P1–N1) values for amplitudes at all tested SPLs of ABR wave I were calculated using ABR Peak Analysis 0.8 RC 1 software (Harvard-MIT Program in Speech and Hearing Bioscience and Technology).

## Immunofluorescence of cochlea frozen section

Cochleae were harvested and fixed in 4% PFA overnight. After decalcification in Decal Stat Decalcifier overnight at 4°C, the cochleae were dehydrated in 15 and 30% sucrose, frozen, and embedded in optimal cutting temperature compound. Sections of the cochleae at 12 µm thickness were cut in the mid-modiolar plane. The specimens were permeabilized in 0.5% Triton X-100 for 30 min, blocked with 10% GS and 1% BSA diluted in 1× PBS for 1 hr at RT, then incubated with the primary antibodies diluted in 10% GS and 1% BSA in 1× PBS overnight at 4°C. After three washes with 1× PBS, the specimens were subsequently stained with fluorescence-conjugated secondary antibodies diluted in 10% GS and 1% BSA in 1× PBS for 1 hr at RT and washed again with 1× PBS three times, then mounted in Antifade Mounting Medium with DAPI. All samples were visualized under an FV1000 Olympus confocal microscope (Olympus FV1000, Japan).

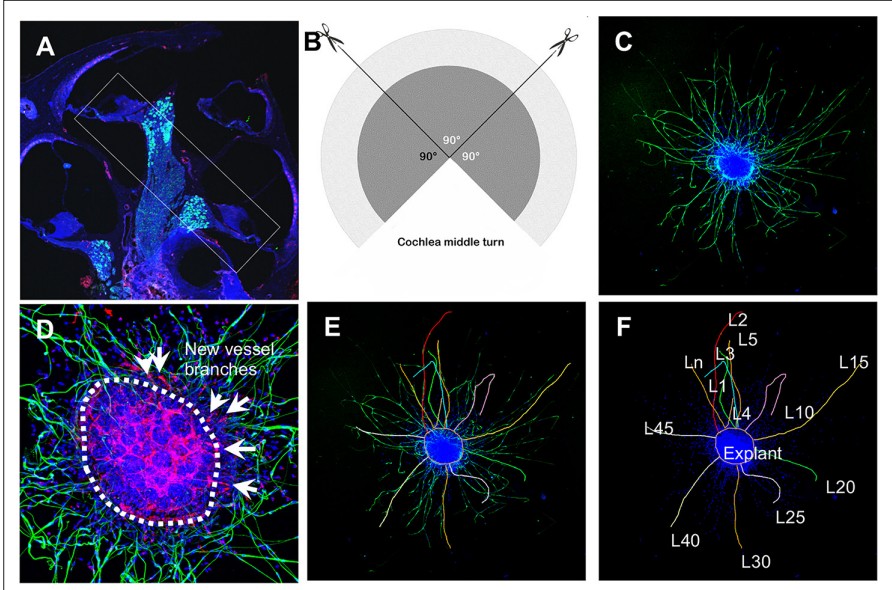

**Figure 12.** Schematic of dissection and culture of spiral ganglion neuron (SGN) explants and method of quantitative analysis of neural dendritic growth in a P2 neonatal mouse cochlea over the course of 5 d in culture. (**A and B**) The cochlear middle turns were dissected out, stria vascularis and organ of Corti discarded, and the remaining SGN cut into three 90° fan-shaped pieces. These were attached to a coated 6-well glass bottom plate and cultured. (**C**) A representative confocal projection image of an SGN explant labeled with an antibody for β-III tubulin (green). (**D–F**) Illustration showing the method used for quantification of SGN neurite number and length. N: the number of neurites; L: the length of the neurites. (**F**) High magnification of the SGN explant showing new vessel growth labeled for CD31 (red). Arrows indicate new vessel branches.

## Assessment of SGN density and intensity

Cochleae cross-sections were stained with β-III Tubulin (1:200). Controls were prepared by replacing primary antibodies with 1% BSA-PBS. Multiple high-resolution images of each cochlea were visualized and acquired under an FV1000 Olympus confocal microscope with a 40× objective. For analysis of SGNs, β-III tubulin labeled cells were visualized and counted over the same measured area of the Rosenthal's canal. All analyses were performed with an image analysis program (Fiji). The average percentage of SGN density (defined as $\frac{SGN\ number}{area\,(10^4\,\mu m^2)}$) and β-III Tubulin intensity for each SGN in three regions (apex, middle, and base) were calculated for each sample.

## SGN explant culture and assessment

SGN explants were isolated from P1 to P3 mouse cochleae of both sexes. To maintain tissue consistency, spiral ganglion tissue from the cochlear middle turn was excised from the entire length of the cochlea and divided into three 90° fan-shaped explants, as illustrated in *Figure 12A and B*, then cultured on a thin gel of Corning Matrigel Growth Factor Reduced Basement Membrane Matrix in SGN basic culture medium consisting of Neurobasal Medium, 1% HEPES, 1% GlutaMAXTM, 2% B-27 Supplement, and 1% Penicillin-Streptomycin Solution (***Note:*** serum-free medium was used to limit the effect of multi growth factors contained in the serum). On the final day in culture, the SGN explants and adult SGNs were fixed with 2% PFA for 30 min on ice. The specimens were permeabilized in 0.5% Triton X-100 for 30 min, blocked with 10% GS and 1% BSA diluted in 1× PBS for 1 hr at RT, then incubated with the antibody for β-III tubulin (1:200) and antibody for CD31 (1:100) diluted in 10% GS and 1% BSA in 1× PBS overnight at 4°C. After three washes with 1× PBS, the specimens were subsequently stained with fluorescence-conjugated secondary antibodies diluted in 10% GS and 1% BSA in 1× PBS for 1 hr at RT, followed by Hoechst 33342 staining for 15 min at RT. All samples were visualized under an FV1000 Olympus confocal microscope (Olympus FV1000, Japan), as shown in *Figure 12C and D*. Using Fiji, the total number of neurites and blood vessel sprouts (new branch formations outside the tissue explants) were counted, then divided by the total area of the explant for quantification. The

average length of the neurites was also assessed, defined as $Average\ neurite\ length = \frac{L1+L2+...+Ln\,(m)}{n}$ (*Figure 12E and F*).

## Adult SGN culture and assessment

Adult SGN cells were isolated and cultured using a modified protocol described by *Vieira et al., 2007*. Briefly, cochleae from 4 wk old mice of both sexes were collected; after the organ of Corti was peeled away, the modiolus was split longitudinally to expose the spiral ganglion and digested in 1 ml Hibernate A containing 2 mg/ml dispase II, 1 mg/ml collagenase I, and 1 mg/ml DNase I for 30 min at 33°C and gently agitated every 5 min. The tissue was dissociated by aspirating and expelling 30 times through a 1 ml polypropylene pipette tip. Tissue clumps were allowed to sediment for 1–2 min, the supernatant was collected and centrifuged at at 80× g for 1 min to remove bony debris. The supernatant was then slowly transferred to a new 15 ml centrifuge tube containing 3 ml of 15% BSA+10% (fetal bovine serum) FBS in Hibernate A as a cushion to remove the fiber debris by centrifuging at 200× g for 10 min. During centrifugation, the cells will pass the cushion and settle in the bottom, while the fiber debris will accumulate in the cushion layer. The pelleted cells were finally resuspended at $3 \times 10^4$ neurons/ml in SGN basic culture medium (see SGN explant culture), and ~3000 SGN cells in 100 µl culture medium were added to each well coated with matrigel matrix diluted in neurobasal medium at a ratio of 1:9. On the final day in culture, the adult SGNs were fixed with 2% PFA for 30 min on ice. The specimens were permeabilized in 0.5% Triton X-100 for 30 min, blocked with 10% GS and 1% BSA diluted in 1× PBS for 1 hr at RT, then incubated with antibodies for β-III tubulin (1:500) diluted in 10% GS and 1% BSA in 1× PBS overnight at 4°C. After three washes with 1× PBS, the specimens were subsequently stained with fluorescence-conjugated secondary antibodies diluted in 10% GS and 1% BSA in 1× PBS for 1 hr at RT, followed by Hoechst 33342 staining for 15 min at RT. All samples were visualized under an FV1000 Olympus confocal microscope (Olympus FV1000, Japan). The number of SGNs per well, neurite number, and longest neurite length of each neuron were quantified using Fiji for statistical analysis.

## Pericyte co-culture with SGN explants and adult SGNs

The primary cochlea pericyte cell line was generated from C57BL/6J mice by a well-established 'mini-chip' protocol previously described and published (*Neng et al., 2013a*), at passage 3–6 used for all experiments. The explants or SGNs were first incubated in 100 µl basic SGN culture medium in glass bottom 6-well plates (P06G-1.5–10 F, MatTek's) overnight at 37°C, 5% $CO_2$, allowing the tissue/cell well to attach to the device. The next day, culture medium was added up to 3 ml, and pericytes at $3.0 \times 10^5$ cells/well were seeded in the insert (Cat#: 353091, Corning) for five continuous days. The explant was first incubated in 100 µl SGN medium in glass bottom 6-well plates overnight at 37°C, 5% $CO_2$. The next day, the culture medium was added up to 3 ml, and pericytes (passage 3–6) at $3.0 \times 10^5$ cells/well were seeded in the insert for five continuous days.

## RNA-seq analysis

Pericyte specimens were sent to Otogenetics Corporation (Norcross, GA, USA) for bulk RNA-seq assay. Briefly, total RNA was extracted from cell pellets using the E.Z.N.A. Total RNA Kit II and the integrity and purity of total RNA were assessed using Agilent Bioanalyzer and OD260/280. 1–2 µg of cDNA was generated using a Clontech SMARTer cDNA kit from 100 ng of total RNA, and adaptors were removed by digestion with RsaI. The resulting cDNA was fragmented using Covaris (Covaris, Inc, Woburn, MA, USA) or Bioruptor, profiled using an Agilent Bioanalyzer, and subjected to Illumina library preparation using NEBNext reagents (Cat#: E6040, New England Biolabs). The quality and size distribution of the Illumina libraries were determined using an Agilent Bioanalyzer2100. The libraries were then submitted for Illumina HiSeq2000 sequencing, used per standard operation. Paired-end 90 or 100 nucleotide reads were generated and checked for data quality using FASTQC (Babraham Institute, Cambridge, UK) and subjected to data analysis using the platform provided by DNAnexus (DNAnexus, Inc, Mountain View, CA, USA) or the platform provided by the Center for Biotechnology and Computational Biology (University of Maryland, College Park, MD, USA) as previously described (*Trapnell et al., 2012*).

## NTA, TEM, and proteomic analysis of pericyte-derived exosomes

NTA, TEM, and proteomic analyses of pericyte-derived exosomes were performed by Alpha Nano Tech. Pericytes were initially cultured in normal media in 100 mm collagen I coated petri dishes

until ~90% confluence, then rinsed with PBS, and transferred to conditioned medium containing 2% Bovine exosome-free FBS (heat inactivated; Cat#: EXO-FBSHI-50A-1, SBI) for 24 hr. Exosomes were purified from the collected media by ultrafiltration followed by size exclusion separation, as shown in *Figure 5A*. Briefly, the conditioned media were loaded into the pre-rinsed ultrafiltration devices (Vivaspin 20) containing a 100 kDa molecular weight cutoff (MWCO) polyethersulfone (PES) membrane and centrifuged at 3000× g for several intervals of 30 min until the final volume reached 500 µl. The exosome fractions were collected with the Izon 35 nm qEV original column and further concentrated using Amicon Ultra 2 100 kDa MWCO centrifugal filter devices.

## Nanoparticle tracking analysis

The NTA of exosomes labeled with Exoglow was performed with a Zetaview Quatt instrument (Particle Metrix, Ammersee, Germany). To label exosomes with Exoglow, 2 µl of exoglow dye was added to 12 µl of reaction buffer, mixed with 14 µl of sample, and incubated for 15 min at RT. Dilutions were made by mixing PBS filtered through a 0.2 µm syringe filter with a corresponding volume of sample. Particle size distribution histograms were recorded in scatter and fluorescent modes.

## Transmission electron microscopy

Copper carbon formvar grids were glow discharged immediately prior to loading with the sample. Sample was processed undiluted. The grid was floated on a 10 µl sample drop for 15 min, washed twice with water by floating on the drop of water for 30 s, and negatively stained with 2% uranyl acetate by floating on a drop of the stain for 30 s. The grid was blot dried with Whatman paper and imaged on a Jeol 1230 electron microscope.

## Proteomic analysis

5 µg of exosomes were dried via vacuum centrifuging. Then the sample was reconstituted in 8 M urea, reduced with dithiothreitol (DTT), alkylated with iodoacetamide, and digested with trypsin overnight. Peptide samples were cleaned using Pierce Peptide Desalting Spin Columns (Cat#: 89852, ThermoFisher) and analyzed in duplicate by liquid chromatography with tandem mass spectrometry (LC-MS/MS) using a Thermo Easy nLC 1200-QExactive HF (Cat#: LC140, ThermoFisher). Proteins were identified and quantified with Proteome Discoverer 2.5 utilizing the Uniprot mouse database appended with a common contaminants database. Further data analysis was conducted in Perseus (Log2 transformation and GOCC term annotation). GO enrichment analysis on these proteins was performed by using the PANTHER classification system, and redundancy in the lists of enriched GO terms was minimized using REVIGO, with the similarity set to 0.5.

## RT-PCR and ELISA

Total RNA of the pericytes was extracted with an RNeasy micro kit, cDNA synthesized with SuperScript IV First-Strand Synthesis kit, used according to the manufacturer's instructions. RT-PCR products (primers designed by using Primer-BLAST to specifically detect VEGFA expression: forward, 5'-GCAGCGACAAGGCAGACTA-3'; reverse, 5'-GGTCCGATGCAAGATCCCAA-3', 392 bp product) were analyzed by 1.5% agarose gel electrophoresis. VEGFA expression level in the supernatant of pericyte culture media and mouse cochleae homogenate was measured by using mouse VEGFA ELISA Kit per the manufacturer's instructions.

## Western blot

Exosomes were purified from the conditioned medium using ultracentrifugation as described by *Breglio et al., 2020*. Briefly, conditioned medium was first centrifuged at 300× g for 10 min at 4°C, then at 10,000× g for 30 min at 4°C. Finally, exosomes were pelleted by ultracentrifuging at 100,000× g for 70 min at 4°C. The exosome-depleted supernatant was further concentrated with a Pierce Protein Concentrator PES (3 K MWCO, Cat#: 88525, ThermoFisher) at 4000× g and 4°C. Both the concentrated non-exosomal supernatant and the purified exosomes were resuspended in RIPA Lysis and Extraction Buffer supplemented with cOmplete Mini EDTA-Free Protease Inhibitor Cocktail (Roche) and vortexed for 1 min. Protein concentration was analyzed with a BCA protein assay kit. The remaining sample was denatured at 95°C for 5 min in 4×Laemmli buffer and subjected to SDS-PAGE using 4–15% Mini-PROTEAN TGX Precast Protein Gels followed by protein transfer to a 0.45 µm pore

(polyvinylidene difluoride) PVDF membrane (Cat#: 88585, ThermoFisher). Membranes were blocked in 3% BSA in TBS with 0.1% Tween-20 (TBS-T). The primary Abs used were VEGFA (1:400) and PDGFRβ (1:5000). HRP-linked secondary Ab: Goat Anti-rabbit IgG (1:10,000). Protein bands were visualized by chemiluminescence using a SuperSignal West Femto Duration Substrate and Q-View Imager System (Quansys Bioscience, Logan, UT, USA).

## Treatment of SGN explants and adult SGNs with pericyte-derived exosomes

Exosomes were purified from the conditioned media using Invitrogen Total Exosome Isolation Reagent (from cell culture media; Cat#: 4478359, ThermoFisher) according to the manufacturer's instructions. The explants or SGNs were cultured in 400 µl SGN basic medium in Nunc Lab-Tek II Chambered Coverglasses (Cat#: 155409, ThermoFisher) with exosomes (5 µg/ml), with or without SU5408 (100 nM for SGN explant and 50 nM for SGNs) for five continuous days. Media was changed every 2 d.

## Statistics

All experiments were designed with proper controls. The sample size estimation was conducted based on our pilot data by power analysis (Sample Size Calculator). To avoid variation and experimental bias, we applied a blind control procedure. All statistical analyses were performed using GraphPad Prism 9 software (GraphPad Software). Statistical difference between two groups was evaluated by unpaired, two-tailed t test. One-way ANOVA followed by Tukey's multiple comparison test was used to compare differences across multiple groups. Two-way ANOVA followed by Dunnett's multiple comparison test was used for ABR analysis to compare threshold or wave I latency differences at different time points and wave I amplitude differences at different SPLs within multiple groups. Differences were considered significant at $p < 0.05$. Data were presented as the mean ± SEM.

## Acknowledgements

Authors sincerely thank Dr. Peter Garr-Gillespie at Oregon Health and Science University, for edits and discussion suggestions on the revised letter of Response, and Dr. Michael Hoa at National Institutes of Health, for the edits and discussion suggestions on the manuscript.

This research was supported by NIH/NIDCD R21 DC016157 (X Shi), NIH/NIDCD R01 DC015781 (X Shi), NIH/NIDCD R01-DC010844 (X Shi).

## Additional information

### Funding

| Funder | Grant reference number | Author |
| --- | --- | --- |
| National Institute on Deafness and Other Communication Disorders | R01 DC015781 | Xiaorui Shi |
| National Institute on Deafness and Other Communication Disorders | R01 DC010844 | Xiaorui Shi |
| National Institute on Deafness and Other Communication Disorders | R21 DC016157 | Xiaorui Shi |

The funders had no role in study design, data collection and interpretation, or the decision to submit the work for publication.

### Author contributions

Yunpei Zhang, Formal analysis, Investigation, Methodology, Validation, Visualization, Writing – original draft, Writing – review and editing; Lingling Neng, Kushal Sharma, Formal analysis, Validation, Investigation, Visualization, Methodology, Writing – original draft; Zhiqiang Hou, Anatasiya Johnson,

Investigation, Methodology; Junha Song, Visualization; Alain Dabdoub, Writing – review and editing; Xiaorui Shi, Resources, Supervision, Funding acquisition, Writing – original draft, Project administration, Writing – review and editing

**Author ORCIDs**
Yunpei Zhang http://orcid.org/0000-0002-5254-796X
Zhiqiang Hou http://orcid.org/0000-0002-5154-7474
Alain Dabdoub http://orcid.org/0000-0002-4259-2425
Xiaorui Shi http://orcid.org/0000-0002-6068-1413

**Ethics**
All animal experiments reported were approved by the Oregon Health & Science University Institutional Animal Care and Use Committee (IACUC IP00000968).

**Decision letter and Author response**
Decision letter https://doi.org/10.7554/eLife.83486.sa1
Author response https://doi.org/10.7554/eLife.83486.sa2

---

## Additional files

**Supplementary files**
• MDAR checklist

**Data availability**
Table 1 - Source Data 1 contains the RNA sequencing data used to generate Table 1 and Figure 5A. Figure 6 - Source Data 1 contains the proteomics data used to generate Figure 6 and Table 2. Figure 7 - Source Data 1 contains the original uncropped blots of VEGFA and PDGFRβ, and raw gel data of whole protein staining with the relevant bands clearly labelled.

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

## Appendix 1

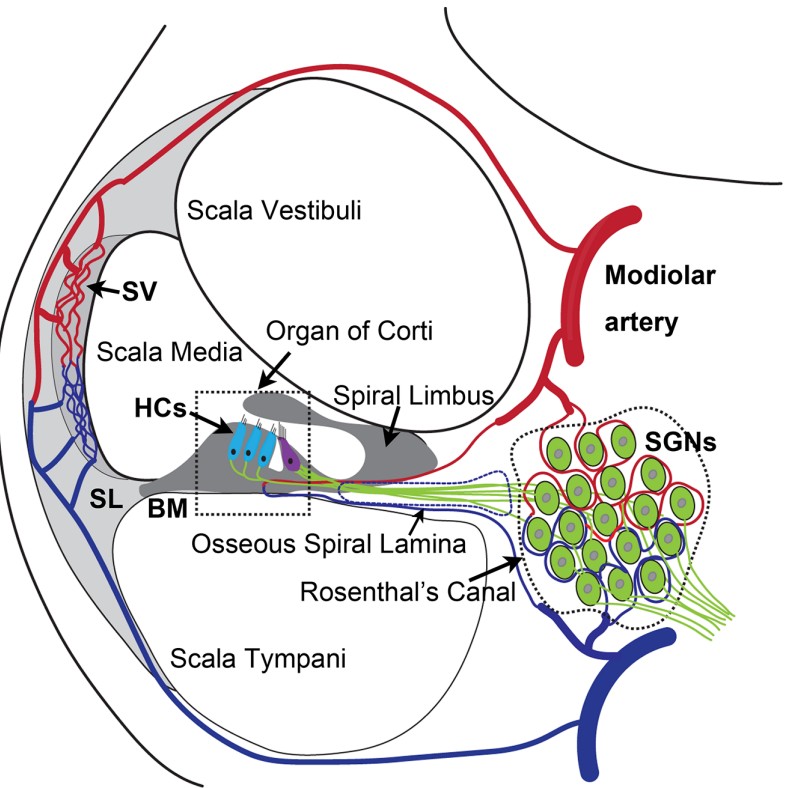

**Appendix 1—figure 1.** The anatomy and blood supply of cochlea. Two major microvascular networks in the cochlea include the network in the cochlear lateral wall and network in the region of spiral ganglion neurons (SGNs)-blood vessels penetrate the SGNs and directly supply nutrients to the neurons. SV, stria vascularis; SL, spiral ligament; HCs, hair cells; BM, basilar membrane.

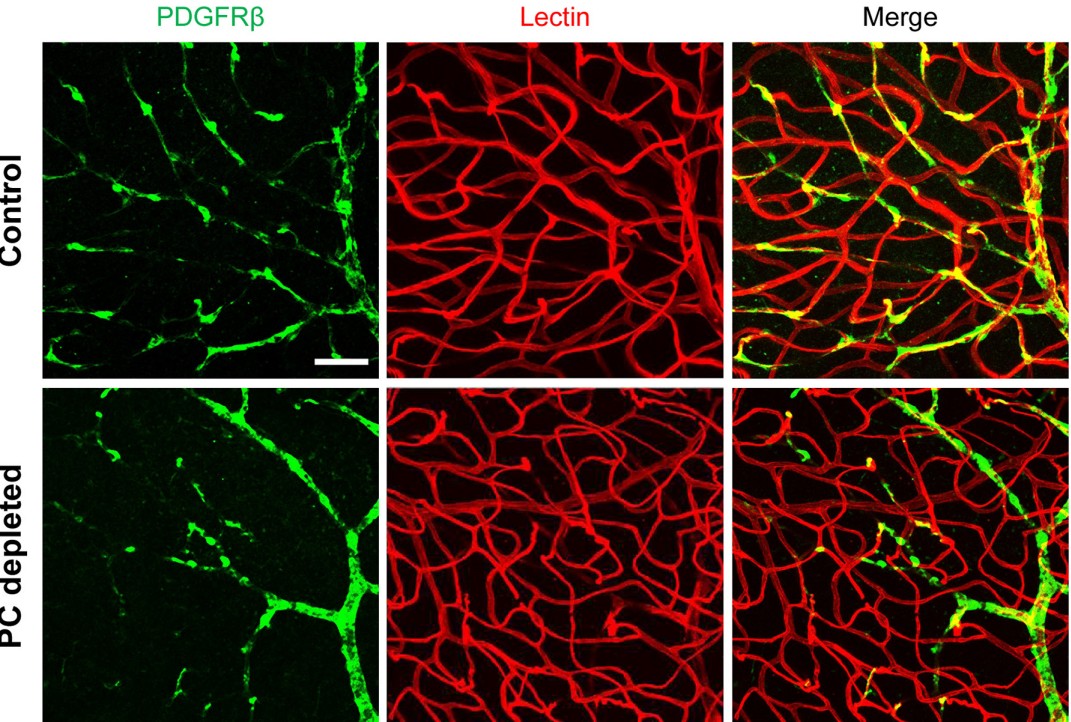

**Appendix 1—figure 2.** Pericyte depletion in the retina does not affect the blood vessels in adult animals. Scale bar: 50 µm.

