## [Editor Report]

This study identifies the roles of the pericytes in maintaining vascular volume and integrity of spiral ganglion neurons (SGNs) in the cochlea, the main hearing organ. It demonstrates that the roles are achieved mainly through the interactions between pericyte-released exosomes containing VEGF-A and VEGFR2-expressing the vessels and SGNs. Understanding the roles of organ-specific pericytes is paramount, making this study timely and significant. The study would be interesting for biomedical biologists working on hearing, blood vessels, signaling, and cell-to-cell interactions.

---

## [Decision Letter]

**Decision letter after peer review:**

Thank you for submitting your article "Pericytes control vascular stability and auditory spiral ganglion neuron survival" for consideration by *eLife*. Your article has been reviewed by 3 peer reviewers, including Gou Young Koh as the Reviewing Editor and Reviewer #1, and the evaluation has been overseen by Barbara Shinn-Cunningham as the Senior Editor.

The authors are required to carefully address all of the comments point-by-point in a data-driven manner or with further analyses. I believe the authors could revise the manuscript successfully given their expertise but please let us know if it takes more than 3 months, and provide the reasons for not implementing the suggested changes if that applies.

*Reviewer #1 (Recommendations for the authors):*

1. Deletion of PDGFRβ+ pericytes using the iDTR system is an artificial approach to understanding the roles of the pericytes in each organ. What are the circumstances of pericyte loss or pericyte drop-out in the cochlea? It would be constructive if the authors show such a loss or dropout in pathophysiologic conditions including diabetic condition, aging, and other hearing loss conditions.

2. The authors are required to distinguish the phenotypes of the pericyte deletion, at least the inner ear versus the outer ear (or any organs as negative and positive controls), to strengthen the findings in the inner ear. The phenotype of the pericyte deletion is organ- and tissue-specific. For example, the study by DY Park et al. (Plastic roles of pericytes in the blood-retinal barrier, Nature Comm. 2017) did not show any alterations in the adult retinal blood vessels in the absence of the PDGFRβ+ pericytes. This paper needs to be cited and discussed in this study.

3. Are there any PDGFRβ+ cells (or fibroblasts) rather than the vascular pericytes in the SGNs? There are more PDGFRβ+ fibroblasts than PDGFRβ+ vascular pericytes in most organs. It is required to provide images supporting the selective distribution of PDGFRβ+ pericytes in the SGNs.

*Reviewer #2 (Recommendations for the authors):*

Several points should be clarified through discussion or supported by additional experiments as listed below:

1. Figure 1 is helpful for readers outside the field, but I don't think such introductory illustrations are appropriate to be shown in regular articles.

2. (Figure 2C, D) "Exclusive co-localization" of tdTomato and PDGFRb is not clear to me. Especially, in areas outside the box in C, they look mostly separated. Better images or accurate interpretations should be presented.

3. (Figure 7A) How did you set the PCR primers? Which isoform did you detect? Please clarify.

4. (Figure 11) Red particles might be PC-derived exosomes as described. Is it difficult to show these particles contain VEGF histologically?

5. Pericytes are depleted systemically, making the interpretation of the data difficult. This point should be discussed as a limitation.

6. Is it appropriate to use PdgfrgCreERT2-/- as controls? The toxicity of Cre, in particular for inducible lines, is well known (Brash et al., Circ Res 2020). Did you exclude non-specific damages to pericytes in PdgfrgCreERT2+/- mice after tamoxifen injection?

7. Functional significance of direct neurotrophic effects of VEGF versus indirect effects mediated by angiogenesis has long been debated. Former effects have been mostly supported by in vitro culture, but not in vivo genetic evidence. Ideally, Flk1 deletion in neurons should be performed. In most cases using neuronal Flk1 CKO, the difference is very small in spite of statistical significance.

8. Why do you think pericytes need exosomes to deliver VEGF? People may think diffusible isoforms or truncation would be feasible. This point should be discussed.

9. Related to the above, how does VEGF in exosomes bind Flk1? Are they once internalized or ruptured around neurons? Please discuss.

*Reviewer #3 (Recommendations for the authors):*

- Images in Figure 1 show increased permeability of vessels in pericyte-ablated mice. As pericytes are important for controlling vessel permeability, it would be important that the authors comment on it.

- It could be that VEGF is not only expressed by pericytes but also but other cell types in the inner ear (i.e. endothelial cells and neurons). Thus, it would be important that the authors complement the analysis of VEGF expression not only via Elisa but also via in situ hybridization or immunohistochemistry.

- In Figure 10 the authors show reduced neurite length when explants are cultured with pericyte conditioned medium. However, is this effect due to VEGF acting directly on neurons or indirectly in endothelial cells? Also of note: the title of Figure 10 is not easy to understand.

- Figure 11 should be presented in the results section and not in the discussion. Also, with the presented images it is not clear what the claim that SGN takes up particles released by pericytes is. Better evidence is recommended.

---

## [Author Response]

Reviewer #1 (Recommendations for the authors):1. Deletion of PDGFRβ+ pericytes using the iDTR system is an artificial approach to understanding the roles of the pericytes in each organ. What are the circumstances of pericyte loss or pericyte drop-out in the cochlea? It would be constructive if the authors show such a loss or dropout in pathophysiologic conditions including diabetic condition, aging, and other hearing loss conditions.

Reviewer #1 makes a significant and insightful point regarding the translation potential of these findings. We agree the iDTR/ PDGFRβ+ depletion animal model is not a disease model, but rather an artificial biological model. The advantage of this model is it allows us to study the direct role of pericytes in cochlear health while avoiding other complications which occur in a disease model. Reduction in pericyte number, migration of pericytes, and pericyte trans-differentiation are well-defined in aging and noise-induced hearing loss, demonstrated in both our early work (Neng et al., 2015) and more recent work (Hou et al., 2020; Hou et al., 2018).

2. The authors are required to distinguish the phenotypes of the pericyte deletion, at least the inner ear versus the outer ear (or any organs as negative and positive controls), to strengthen the findings in the inner ear. The phenotype of the pericyte deletion is organ- and tissue-specific. For example, the study by DY Park et al. (Plastic roles of pericytes in the blood-retinal barrier, Nature Comm. 2017) did not show any alterations in the adult retinal blood vessels in the absence of the PDGFRβ+ pericytes. This paper needs to be cited and discussed in this study.

Yes, pericytes are organ-oriented and tissue specific. In the revision, we have added supplemental data demonstrating that while pericyte depletion in our model affects retinal pericytes, we have not observed obvious abnormalities of retina blood vessels in adult animals (Appendix-Figure 2), consistant with the previous report by DY Park et al. (2017). We also added discussion on the effect of pericyte depletion in different organs on page 7, lines 286-297.

3. Are there any PDGFRβ+ cells (or fibroblasts) rather than the vascular pericytes in the SGNs? There are more PDGFRβ+ fibroblasts than PDGFRβ+ vascular pericytes in most organs. It is required to provide images supporting the selective distribution of PDGFRβ+ pericytes in the SGNs.

Great Question! Yes, PDGFRβ can be expressed cells other than pericytes, including fibroblasts and glia (Hewitt et al., 2012). In the cochlea, we find PDGFRβ is mostly expressed by pericytes. However, there is a small population of PDGFRβ positive cells in non-vascular regions that are not pericytes. The population is small and insignificant, however, compared to the majority of PDGFRβ+ pericytes. We have now provided better images on the distribution of PDGFRβ+ pericytes in the spiral ganglion region (see Figures2C and D).

Reviewer #2 (Recommendations for the authors):Several points should be clarified through discussion or supported by additional experiments as listed below:1. Figure 1 is helpful for readers outside the field, but I don't think such introductory illustrations are appropriate to be shown in regular articles.

Thanks for the suggestion. We moved Figure 1 to appendix 1- Figure 1 to better inform non-auditory scientists on the location of vascular structure in the cochlea.

2. (Figure 2C, D) "Exclusive co-localization" of tdTomato and PDGFRb is not clear to me. Especially, in areas outside the box in C, they look mostly separated. Better images or accurate interpretations should be presented.

We apologize for the definitive statement. We have revised the text to accurately reflect the findings. Reviewer #1 might agree that immunostaining in the area is challenging due to the complexity of the tissue structure. We have replaced the original image with better images (see Figures 2C and D). But, overall, the Reviewer is correct. While some PDGFRβ+ cells are outside of blood vessels and are of unknown cell type, most PDGFRβ+ cells are vascular perciytes. This could be due to that the underside of vessels is not captured because of the thickness of the spiral lamina. Visualization of vascular structure in large-scale volumetric models in this region is often difficult and limited by the optical scattering and absorption of light by bone. In our 2019 publication (Jiang et al., 2019), we showed that details of vascular architecture in the spiral limbus and SGN in whole-mounted mouse preparations are not fully revealed without tissue clearing. However, when we do use tissue clearance, the procedure damages membrane proteins and immunolabeling for PDGFRβ fails. In addition, some of the tdTomato-positive structures apparently not co-labeled by PDGFRβ antibodies could also be the result of a common signal-leakage side-effect in transgenic mouse models (Song and Palmiter, 2018; Stifter and Greter, 2020). Nevertheless, since we only deplete partial of the PDGFRβ cells in the vessel beds, we believe the small number of ‘non-pericytes’ is insignificant.

3. (Figure 7A) How did you set the PCR primers? Which isoform did you detect? Please clarify.

Thanks for the question. We designed the PCR primers using Primer-BLAST to specifically detect for VEGFA expression. Details have been added to the Methods and Materials section on page 16, lines 608-610.

4. (Figure 11) Red particles might be PC-derived exosomes as described. Is it difficult to show these particles contain VEGF histologically?

We apologize for the confusion; our description of Figure 11 was unclear. The purpose of Figure 11 in the initial submission was to demonstrate active communication between pericytes and the spiral ganglion *(*material from one cell interacting with another cell). We are not certain what those red particles are. They could be particles released by pericytes. But it is very difficult to identify the content of exosomes without high-resolution EM combined with immunochemical binding for VEGFA. The current image resolution used is by far insufficient to visualize exosomes and their contents. To provide the reader with on this issue, and follow the Reviewer’s suggestion, we moved Figure 11 to Figure 1. The original Figure 11 has also been replaced with a better data set to demonstrate pericyte-SGN cellular communication.

5. Pericytes are depleted systemically, making the interpretation of the data difficult. This point should be discussed as a limitation.

Thanks to reviewer #2 for the comments. We have added discussion of the limitations of our study to the Discussion section on page 7, lines 284-297.

6. Is it appropriate to use PdgfrgCreERT2-/- as controls? The toxicity of Cre, in particular for inducible lines, is well known (Brash et al., Circ Res 2020). Did you exclude non-specific damages to pericytes in PdgfrgCreERT2+/- mice after tamoxifen injection?

Great point. We used PdgfrbCreERT2-/- mice as controls to test the toxicity of both tamoxifen and DT in PdgfrbCreERT2-/-;DTR+/- mice. To test the toxicity of Cre, we assessed the ABR in PdgfrbCreERT2+/-;DTR+/- mice after tamoxifen injection and before injection of DT. We did not observe any ABR threshold difference between the PdgfrbCreERT2-/- and PdgfrbCreERT2+/- mice (Figure 3A). Therefore the dose of tamoxifen we used did not affect hearing function.

7. Functional significance of direct neurotrophic effects of VEGF versus indirect effects mediated by angiogenesis has long been debated. Former effects have been mostly supported by in vitro culture, but not in vivo genetic evidence. Ideally, Flk1 deletion in neurons should be performed. In most cases using neuronal Flk1 CKO, the difference is very small in spite of statistical significance.

Yes, we fully agree! Our study has shown Flk1 strongly expressed in peripheral auditory neurons (as shown in Figures8C and D) and in isolated adult SGNs. Our in vitro data also demonstrated SGNs directly respond to VEGF-A containing vesicles, as shown in Figure 9C. The exosome-treated adult SGNs showed more SGN dendritic growth (red, labeled for β-III tubulin) relative to control and VEGFR2-inhibited groups. Taken together the data indicates the direct effect of VEGF-A in SGN growth. However, it would be ideal if further evidence could be provided in a conditional SGN-Flk1 KO mouse model, as the Reviewer has suggested. We have mentioned this in the Discussion on page 9, lines 403-406.

8. Why do you think pericytes need exosomes to deliver VEGF? People may think diffusible isoforms or truncation would be feasible. This point should be discussed.

Great point! Yes, we agree that VEGF-A is directly released by cells in diffusible isoforms. However, in our early study, we observed that cochlear pericytes release particles, and cellular communication between pericytes and SGNs in vivo*,* as shown in the new Figure 1. In vitro PC culture medium is rich in exosomes. With protein immunoblots, we have determined VEGF-A is predominant in the exosomes, suggesting the VEGF-A is delivered by exosomes. That said, why pericytes need to deliver VEGF-A in exosomes is not clear. Studies in other organ systems do show shedding of VEGFA-165 from the cell surface, together with other membrane components; this appears to be a vehicle by which some VEGF is delivered to the surroundings to exert its known biological actions (Guzman-Hernandez et al., 2014). However, the exact mechanism by which pericytes release VEGFA in the form of exosomes and affect SGN health is unknown and needs further investigation. Thanks to the reviewer for pointing this out. We have mentioned this in the Discussion on page 9, lines 372-379.

9. Related to the above, how does VEGF in exosomes bind Flk1? Are they once internalized or ruptured around neurons? Please discuss.

This is an excellent question, one we asked ourselves during the study. In general, exosome-driven mechanisms are complicated and poorly understood, which we have pointed out in a recent review article (Sharma et al., 2022). For example, studies demonstrate that EVs interface with neighboring cells through different modes, including direct receptor-binding without internalization of EVs, phagocytosis, macropinocytosis, internalization by clathrin-caveolae- and lipid raft-mediated endocytosis (Mulcahy et al., 2014). Once internalized, the EVs release their cargo in the cytoplasm and the cell-internalized cargo then regulates the cell at the transcription or translation level (Abels & Breakefield, 2016; Berumen Sanchez et al., 2021). VEGF also presents on the surface of EVs by binding to surface proteins such as heparin and heat shock protein 90 (HSP90), which can directly stimulate the cellular domain of Flk1 and induce phosphorylation (Ko et al., 2019; Ko & Naora, 2020). We have now added discussion of these mechanisms on page 9, lines 379-388. It is our future goal to identify the underlying mechanisms in collaboration with cell biologists with expertise in this area of research.

Reviewer #3 (Recommendations for the authors):- Images in Figure 1 show increased permeability of vessels in pericyte-ablated mice. As pericytes are important for controlling vessel permeability, it would be important that the authors comment on it.

Great point! Yes, pericytes play a significant role in blood vessel integrity, particularly in the blood labyrinth and blood-brain barriers, as we showed in our early studies (Hou et al., 2018; Zhang et al., 2021). Pericytes stabilize vascular integrity by releasing signaling factors that control the expression of tight junction proteins in endothelial cells. Without pericytes, tight junction protein is significantly downregulated in vascular systems of the cochlea (Neng et al., 2013). We have now added comments on this in the Discussion section on page 8, lines 328-334.

- It could be that VEGF is not only expressed by pericytes but also but other cell types in the inner ear (i.e. endothelial cells and neurons). Thus, it would be important that the authors complement the analysis of VEGF expression not only via Elisa but also via in situ hybridization or immunohistochemistry.

Yes, the reviewer is correct. Earlier studies by Pasqualina et al., 2004 demonstrated that VEGF is expressed in supporting cells, hair cells, stria vascularis, and spiral ganglia (Picciotti et al., 2004). It is likely that VEGF signaling in these cells in part plays a role in regulation of cochlear and hearing function. Our study has shown for the first time that VEGF is expressed in pericytes and secretion of VEGF participates in regulation of vascular and neuronal function. In accord with Reviewer#3’s suggestion, we added new images of the VEGF immunostaining to demonstrate that VEGFA is expressed in pericytes (Fig.7D). We appreciate the reviewer’s suggestion to conduct an in situ hybridization, which could indeed confirm mRNA expression for VEGF in the pericytes. However, since our RNA seq data already show expression of VEGF mRNA in pericytes, the in situ hybridization would be redundant; the new immunostaining data should therefor be sufficient for demonstrating expression in pericytes.

- In Figure 10 the authors show reduced neurite length when explants are cultured with pericyte conditioned medium. However, is this effect due to VEGF acting directly on neurons or indirectly in endothelial cells? Also of note: the title of Figure 10 is not easy to understand.

Thanks for pointing out this problem. To ensure that we are assessing the direct effect of VEGF in neurons, we treated isolated adult neurons with pericyte-derived exosomes with and without the specific VEGFR2 inhibitor, SU5408, as shown in Figs. 9C & E. This method eliminated the possible neighborhood effects of endothelial and supporting cells. Fig. 10 shows results from a preliminary experiment to optimize the dose of the VEGFR2 inhibitor in SGN explants. The title of Fig. 10 has been changed for clarity.

- Figure 11 should be presented in the results section and not in the discussion. Also, with the presented images it is not clear what the claim that SGN takes up particles released by pericytes is. Better evidence is recommended.

Great suggestion. In accord with the Reviewer’s suggestion, we have moved Fig.11 to the Results section and redubbed it Fig.1. The original Fig.11 is also replaced with a better image. The purpose of this Figure is to demonstrate cellular communication between pericytes and SGNs in vivo. We have correspondingly revised our interpretation in the Results section and Figure legend.

References

Abels, E. R., and Breakefield, X. O. (2016). Introduction to Extracellular Vesicles: Biogenesis, RNA Cargo Selection, Content, Release, and Uptake. *Cell Mol Neurobiol*, *36*(3), 301-312. https://doi.org/10.1007/s10571-016-0366-z

Berumen Sanchez, G., Bunn, K. E., Pua, H. H., and Rafat, M. (2021). Extracellular vesicles: mediators of intercellular communication in tissue injury and disease. *Cell Commun Signal*, *19*(1), 104. https://doi.org/10.1186/s12964-021-00787-y

Deyama, S., Li, X. Y., and Duman, R. S. (2020). Neuron-specific deletion of VEGF or its receptor Flk-1 impairs recognition memory. *Eur Neuropsychopharmacol*, *31*, 145-151. https://doi.org/10.1016/j.euroneuro.2019.11.002

Guzman-Hernandez, M. L., Potter, G., Egervari, K., Kiss, J. Z., and Balla, T. (2014). Secretion of VEGF-165 has unique characteristics, including shedding from the plasma membrane. *Mol Biol Cell*, *25*(7), 1061-1072. https://doi.org/10.1091/mbc.E13-07-0418

Hewitt, K. J., Shamis, Y., Knight, E., Smith, A., Maione, A., Alt-Holland, A., Sheridan, S. D., Haggarty, S. J., and Garlick, J. A. (2012). PDGFRbeta expression and function in fibroblasts derived from pluripotent cells is linked to DNA demethylation. *J Cell Sci*, *125*(Pt 9), 2276-2287. https://doi.org/10.1242/jcs.099192

Hou, Z., Neng, L., Zhang, J., Cai, J., Wang, X., Zhang, Y., Lopez, I. A., and Shi, X. (2020). Acoustic Trauma Causes Cochlear Pericyte-to-Myofibroblast-Like Cell Transformation and Vascular Degeneration, and Transplantation of New Pericytes Prevents Vascular Atrophy. *Am J Pathol*, *190*(9), 1943-1959. https://doi.org/10.1016/j.ajpath.2020.05.019

Hou, Z., Wang, X., Cai, J., Zhang, J., Hassan, A., Auer, M., and Shi, X. (2018). Platelet-Derived Growth Factor Subunit B Signaling Promotes Pericyte Migration in Response to Loud Sound in the Cochlear Stria Vascularis. *J Assoc Res Otolaryngol*, *19*(4), 363-379. https://doi.org/10.1007/s10162-018-0670-z

Jiang, H., Wang, X., Zhang, J., Kachelmeier, A., Lopez, I. A., and Shi, X. (2019). Microvascular networks in the area of the auditory peripheral nervous system. *Hear Res*, *371*, 105-116. https://doi.org/10.1016/j.heares.2018.11.012

Ko, S. Y., Lee, W., Kenny, H. A., Dang, L. H., Ellis, L. M., Jonasch, E., Lengyel, E., and Naora, H. (2019). Cancer-derived small extracellular vesicles promote angiogenesis by heparin-bound, bevacizumab-insensitive VEGF, independent of vesicle uptake. *Commun Biol*, *2*, 386. https://doi.org/10.1038/s42003-019-0609-x

Ko, S. Y., and Naora, H. (2020). Extracellular Vesicle Membrane-Associated Proteins: Emerging Roles in Tumor Angiogenesis and Anti-Angiogenesis Therapy Resistance. *Int J Mol Sci*, *21*(15). https://doi.org/10.3390/ijms21155418

Mulcahy, L. A., Pink, R. C., and Carter, D. R. (2014). Routes and mechanisms of extracellular vesicle uptake. *J Extracell Vesicles*, *3*. https://doi.org/10.3402/jev.v3.24641

Neng, L., Zhang, F., Kachelmeier, A., and Shi, X. (2013). Endothelial cell, pericyte, and perivascular resident macrophage-type melanocyte interactions regulate cochlear intrastrial fluid-blood barrier permeability. *J Assoc Res Otolaryngol*, *14*(2), 175-185. https://doi.org/10.1007/s10162-012-0365-9

Neng, L., Zhang, J., Yang, J., Zhang, F., Lopez, I. A., Dong, M., and Shi, X. (2015). Structural changes in thestrial blood-labyrinth barrier of aged C57BL/6 mice. *Cell Tissue Res*, *361*(3), 685-696. https://doi.org/10.1007/s00441-015-2147-2

Picciotti, P., Torsello, A., Wolf, F. I., Paludetti, G., Gaetani, E., and Pola, R. (2004). Age-dependent modifications of expression level of VEGF and its receptors in the inner ear. *Exp Gerontol*, *39*(8), 1253-1258. https://doi.org/10.1016/j.exger.2004.06.003

Sharma, K., Zhang, Y., Paudel, K. R., Kachelmeier, A., Hansbro, P. M., and Shi, X. (2022). The Emerging Role of Pericyte-Derived Extracellular Vesicles in Vascular and Neurological Health. *Cells*, *11*(19). https://doi.org/10.3390/cells11193108

Song, A. J., and Palmiter, R. D. (2018). Detecting and Avoiding Problems When Using the Cre-lox System. *Trends Genet*, *34*(5), 333-340. https://doi.org/10.1016/j.tig.2017.12.008

Stifter, S. A., and Greter, M. (2020). STOP floxing around: Specificity and leakiness of inducible Cre/loxP systems. *Eur J Immunol*, *50*(3), 338-341. https://doi.org/10.1002/eji.202048546

Zhang, J., Hou, Z., Wang, X., Jiang, H., Neng, L., Zhang, Y., Yu, Q., Burwood, G., Song, J., Auer, M., Fridberger, A., Hoa, M., and Shi, X. (2021). VEGFA165 gene therapy ameliorates blood-labyrinth barrier breakdown and hearing loss. *JCI Insight*, *6*(8). https://doi.org/10.1172/jci.insight.143285